# CONTINUITY-PRESERVING CONVOLUTIONAL AUTOENCODERS FOR LEARNING CONTINUOUS LATENT DYNAMICAL MODELS FROM IMAGES

**Aiqing Zhu**[1], **Yuting Pan**[2,3], **Qianxiao Li**[1,4*]
[1]Department of Mathematics, National University of Singapore
[2]School of Computing, National University of Singapore    [3]CNRS@CREATE LTD
[4]Institute for Functional Intelligent Materials, National University of Singapore
`zaq@nus.edu.sg`, `pan.yuting@u.nus.edu`, `qianxiao@nus.edu.sg`

## ABSTRACT

Continuous dynamical systems are cornerstones of many scientific and engineering disciplines. While machine learning offers powerful tools to model these systems from trajectory data, challenges arise when these trajectories are captured as images, resulting in pixel-level observations that are discrete in nature. Consequently, a naive application of a convolutional autoencoder can result in latent coordinates that are discontinuous in time. To resolve this, we propose continuity-preserving convolutional autoencoders (CpAEs) to learn continuous latent states and their corresponding continuous latent dynamical models from discrete image frames. We present a mathematical formulation for learning dynamics from image frames, which illustrates issues with previous approaches and motivates our methodology based on promoting the continuity of convolution filters, thereby preserving the continuity of the latent states. This approach enables CpAEs to produce latent states that evolve continuously with the underlying dynamics, leading to more accurate latent dynamical models. Extensive experiments across various scenarios demonstrate the effectiveness of CpAEs.

## 1 INTRODUCTION

Continuous dynamical systems, described by differential equations, are widely used as scientific modeling tools across various biological, physical, and chemical processes. While traditionally described by mathematical models, the increasing availability of data has spurred the development of data-driven approaches (Brunton et al., 2016; Brunton & Kutz, 2022; Schmidt & Lipson, 2009). In particular, machine learning and neural networks have recently emerged as powerful tools, achieving remarkable success in tasks such as discovering (Chen et al., 2018; González-García et al., 1998; Raissi et al., 2018), predicting (Wang et al., 2021; Wu & Xiu, 2020; Xie et al., 2024), and controlling (Brunton & Kutz, 2022; Chen et al., 2023; Zhong et al., 2020) continuous dynamical systems based on observed data.

Most methods for modeling dynamical systems are designed for observed data that already correspond to relevant state variables. However, in many scientific and engineering applications, we only have access to measurements that yield a series of discrete image data (Botev et al., 2021; Chen et al., 2022). When applied to image data, a common approach involves using autoencoders to encode natural images located in high-dimensional pixel space onto a low-dimensional manifold (Greydanus et al., 2019; Jin et al., 2023; Toth et al., 2020). It is then assumed that the encoded sequences follow continuous paths governed by a differential equation on this manifold, and machine learning methods automatically capture the continuous dynamics (Botev et al., 2021; Toth et al., 2020). While this approach and assumption have been substantiated on relatively simple tasks, complex visual patterns and dynamical behaviors remain challenging (Botev et al., 2021).

The continuous evolution of dynamical systems over time is a fundamental characteristic of many fields. Machine learning methods for capturing continuous dynamics from discrete data along con-

---

*Corresponding author.

tinuous trajectories have advanced significantly, with robust, general-purpose algorithms now readily available (Chen et al., 2018; Krishnapriyan et al., 2023; Ott et al., 2021). However, image data pose a key challenge since pixel coordinates often do not align with the continuous evolution of underlying dynamics, as illustrated in Fig. 1. Whereas latent states that evolve continuously over time are essential for discovering continuous latent dynamical systems, standard autoencoders often struggle to learn such valuable latent representations, as will be discussed in detail later.

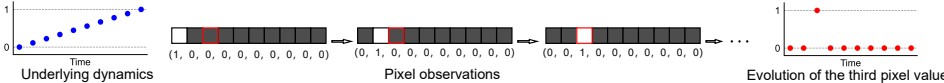

Figure 1: Illustration of pixel observations of continuous motion. A single pixel white square, initially located at the leftmost position, moves uniformly to the right against a black background (plotted in gray for clarity). Its position is recorded at each pixel step. This translational motion results in pixel coordinates that first increase by one, then decrease by one, and finally remain constant. As an illustrative example, we show the evolution of the value at the third pixel position.

We introduce continuity-preserving convolutional autoencoders (CpAEs) to learn continuously evolving latent states from discrete image frames. Our contributions are summarized as follows:

- We propose a mathematical formulation for learning continuous dynamics from image data to describe the continuity of latent states.
- We establish a sufficient condition (Theorem 3.1), demonstrating that the latent states will evolve continuously with the underlying dynamics if the filters are Lipschitz continuous.
- We introduce a regularizer (Eq. (6)) to promote the continuity of filters and, consequently, preserve the continuity of the latent states.
- We perform several experiments across various scenarios to verify the effectiveness of the proposed method.

## 1.1 RELATED WORKS

**Deep Autoencoder**. An autoencoder (Baldi, 2012; Ranzato et al., 2007; Rumelhart et al., 1985) is a type of neural network that encodes input data into a compressed and meaningful representation, then decodes it to reconstruct the original input. A significant advancement in this area has been the development of Variational Autoencoders (VAEs) (Kingma & Welling, 2014), which extend traditional autoencoders by incorporating probabilistic modeling of the latent space. Autoencoders can be adapted and extended to various models, finding broad applications, including image generation and classification (Kingma & Welling, 2014; Pu et al., 2016), data clustering (Guo et al., 2017; Song et al., 2013), and anomaly detection (Gong et al., 2019; Zong et al., 2018). In this paper, we focus on using autoencoders to learn latent dynamical models from images, and we propose novel continuity-preserving autoencoders that incorporate continuity prior for this task.

**Discovering Dynamical model**. Discovering dynamical models from observed time-series data $x_0, x_1, \cdots, x_N$ is a fundamental challenge in science. One effective strategy involves constructing a neural network model, denoted as $\mathcal{N}$, to learn a data-driven flow map that predicts the system's future states (Chen et al., 2022; Chen & Xiu, 2024; Wang et al., 2021; Wu & Xiu, 2020). This model predicts the subsequent state $x_{n+1}$ based on the current state $x_n$. Alternatively, some researchers focus on modeling the governing function of the unknown differential equation (Chen et al., 2018; González-García et al., 1998; Raissi et al., 2018). Given an input $x_n$, $x_{n+1}$ is obtained by solving the NN-parameterized ODE at time $\Delta t$, starting from the initial condition $x_n$. This approach can offer valuable insights into the system's dynamics. For instance, it enables the characterization of invariant distributions (Gu et al., 2023; Lin et al., 2023), energy landscapes (Chen et al., 2023), and other essential properties (Qiu et al., 2022), thereby expanding the scope of scientific investigation.

**Learning dynamics from image observations**. Numerous studies have explored the incorporation of classical mechanics-inspired inductive biases, such as physical principles (Cranmer et al., 2020; Greydanus et al., 2019; Lutter et al., 2019; Yu et al., 2021; Zhang et al., 2022), geometry structures (Eldred et al., 2024; Jin et al., 2020; Zhu et al., 2022), and symmetries (Huh et al., 2020; Yang et al., 2024), into deep neural networks. While some of these models have been applied to learn dynamics from images using autoencoders, they have mostly been tested on relatively simple visual

patterns and dynamical behaviors. An enhanced approach (Botev et al., 2021; Toth et al., 2020) involves employing VAEs to embed images, often resulting in improved predictive and generative performance. Given our priority on ensuring the continuous evolution of the latent states, this work focuses on deterministic autoencoders.

## 2 LEARNING LATENT DYNAMICAL MODELS USING AUTOENCODERS

This paper focuses on learning continuous latent dynamical models from sequential image observations of an unknown system. Our dataset consists of discrete image frames sampled from multiple continuous trajectories, denoted as:

$$\{(X_0^1, X_1^1, \cdots, X_N^1), \cdots, (X_0^M, X_1^M, \cdots, X_N^M)\}, \ X_n^m \in \mathbb{R}^{(I+1) \times (I+1)}, \tag{1}$$

where the superscript indicates the $m$-th trajectory and the subscript indicates the $n$-th time step of the trajectory. We assume that an underlying dynamical system governs the observed image time series, with its governing equations defined as follows:

$$\dot{z} = f(z), \quad z \in \mathcal{Z} \subset \mathbb{R}^D, \tag{2}$$

where $\mathcal{Z}$ denotes the set of states associated with the image observations, and assume that there is a corresponding mapping $\boldsymbol{I}$ from the physical state space to the pixel space. Mathematically, the pixel observation at time $t_n$ is assumed to satisfy $X_n^m = \boldsymbol{I}(z_n^m)$, where $z_n^m$ is the state evolved from initial state $z_0^m$ according to Eq. (2). The true governing function $f$, the underlying physical states $z_n^m$, and the mapping $\boldsymbol{I}$ are all unknown. We further assume that the unknown governing function $f$ is Lipschitz continuous and bounded by $M_f$.

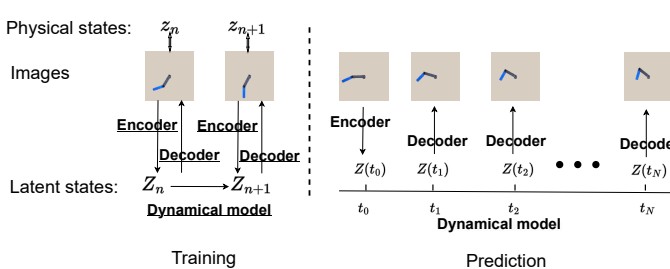

Figure 2: Schematic of learning latent dynamics. The encoder takes images as inputs and infers the corresponding latent states; the decoder maps the latent states back to reconstruct the original images; the dynamical model outputs the subsequent latent state $Z_{n+1}$ based on the current state $Z_n$.

A typical framework for this task comprises three components: an encoder, a dynamical model, and a decoder. An illustration of this framework is shown in Fig. 2. The goals are to: 1) learn an encoder that extract latent states consistent with the assumed latent dynamical system, 2) discover a dynamical model that accurately captures the underlying latent dynamics, and 3) identify a decoder capable of reconstructing the pixel observations.

The first objective is a prerequisite for the entire task; without it, no target dynamical system with assumed structures can be identified. Specifically, to learn a meaningful continuous latent dynamical model, it is essential that the extracted latent states are discrete samples of Lipschitz continuous trajectories. However, without constraints on the encoder, the extracted latent states may deviate from the assumed latent dynamical system. In this paper, we focus on preserving the fundamental continuity of the latent dynamical system. The encoder is carefully designed to ensure the continuous evolution of latent variables over time in this paper.

## 3 CONTINUITY-PRESERVING AUTOENCODER

### 3.1 MATHEMATICAL FORMULATION

Given the discrete nature of image data in both space (pixels) and time, traditional definitions of continuity are not directly applicable. In this section, we propose a mathematical formulation to describe the continuity of latent states when learning continuous dynamics from images.

The formal mathematical definition of projecting the states of a dynamical system into images can be expressed in two steps:

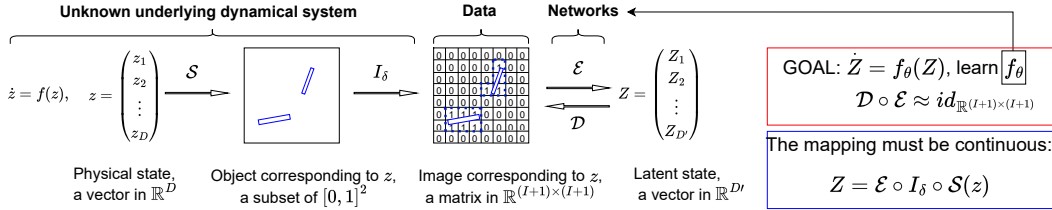

Figure 3: Illustration of the mathematical formulation. The continuous dynamics of the system are captured in pixel form. We learn latent dynamical models by encoding this recorded pixel data.

- **Mapping a state to a set of positions of all the particles constituting the objects**: This mapping is represented as $\mathcal{S}(z) : \mathbb{R}^D \to \mathbf{B}([0,1]^2)$, where $\mathbf{B}([0,1]^2)$ denotes the set of all Borel sets in $[0,1]^2$, $\mathcal{S}(z) =: \Omega$ is the set of positions of particles constituting the objects.
- **Discretizing the coordinate space into image signals, simplifying the images by setting background pixels to 0 and pixels containing objects to 1**: Denote $\delta$ as the pixel size and $(I+1)\delta = 1$. Then we define the functional $\boldsymbol{I}_\delta : \mathbf{B}([0,1]^2) \to \mathbb{R}^{(I+1)\times(I+1)}$ as follows:

$$[\boldsymbol{I}_\delta(\Omega)]_{i_1,i_2} = \begin{cases} 1, & \text{if } [i_1\delta, (i_1+1)\delta] \times [i_2\delta, (i_2+1)\delta] \cap \Omega \neq \emptyset, \\ 0 & \text{otherwise}, \end{cases}$$

where $\Omega$ is a Borel set in $[0,1]^2$, $[\boldsymbol{I}_\delta(\Omega)]_{i_1,i_2}$ denotes the element located at the $(I+1-i_2)$-th row and $(i_1+1)$-th column of the matrix $\boldsymbol{I}_\delta(\Omega) \in \mathbb{R}^{(I+1)\times(I+1)}$.

Finally, we define the observed image data for any state $z$ as $\mathcal{I}_\delta \circ \mathcal{S}(z)$. See Fig. 3 for an illustration.

To reduce dimensionality, CNN typically employ a parameter called *stride* to progressively down-sample feature maps. It is well-known that the convolution operation for two-dimensional input images can be expressed as follows (Goodfellow et al., 2016):

Input: $\qquad \boldsymbol{I}_0 \in \mathbb{R}^{(I+1)\times(I+1)},$

Hidden layers: $\quad [\boldsymbol{I}_l]_{i_1,i_2} = \sum_{j_1=0}^{J_l} \sum_{j_2=0}^{J_l} [\boldsymbol{I}_{l-1}]_{s_l \cdot i_1 + j_1, s_l \cdot i_2 + j_2} \cdot [\boldsymbol{W}_l]_{j_1,j_2}, \ l = 1, \cdots, L,$ (3)

Output: $\qquad \mathcal{E}(\boldsymbol{I}_0) = \boldsymbol{I}_L.$

Here $s_l$ is the stride of the $l$-th layer; $\boldsymbol{I}_l \in \mathbb{R}^{(\lfloor I/\prod_{i=1}^l s_i \rfloor + 1)\times(\lfloor I/\prod_{i=1}^l s_i \rfloor + 1)}$ is the output feature map of the $l$-th layer, and we assume that $\|\boldsymbol{I}_l\| = \mathcal{O}(1)$; $\boldsymbol{W}_l \in \mathbb{R}^{(J_l+1)\times(J_l+1)}$ is the filter of $l$-th layer. For simplicity, we have omitted operations that do not significantly affect continuity, such as activation layers. Additionally, to simplify index counting, we assume that zero padding is applied only on one side of the feature maps, meaning that $[\boldsymbol{I}_l]_{i_1,i_2} = 0$ if $i_1 \vee i_2 \in \lfloor I/\prod_{i=1}^l s_i \rfloor + \{1, 2, \cdots, J_l\}$.

Different resolutions $\delta = 1/(I+1)$ correspond to specific networks, where hyper-parameters such as $J_l$ should be adjusted accordingly. Therefore, we will also denote the feature map as $\boldsymbol{I}_l^\delta$, the filter as $\boldsymbol{W}_l^\delta$ and the CNN as $\mathcal{E}_\delta$ to emphasize their dependence on the pixel size. For the convenience of analysis, we assume the weights $\boldsymbol{W}_l^\delta$ of a standard CNN filter can be normalized to a bounded function $\mathcal{W}_l : \mathbb{R}^2 \to [-1, 1]$ that is independent of $\delta$:

$$[\boldsymbol{W}_l^\delta]_{j_1,j_2} = \mathcal{W}_l(j_1\delta, j_2\delta)\varepsilon_l, \quad \text{where } \mathcal{W}_l(x) = 0 \text{ if } x \notin [0, J_l\delta]^2.$$

Here $\varepsilon_l$ is a normalization coefficient to ensure that $\|\boldsymbol{I}_l\| = \mathcal{O}(1)$ and the last equation ensures that the filter function outputs 0 when the index exceeds the defined size limits.

Now we are ready to provide the following definition of continuity, which serves as a relaxation of the traditional Lipschitz continuity.

**Definition 3.1.** *A sequence of functions $\{g_\delta(z) : \mathcal{Z} \to \mathbb{R}^d | \delta \in \{1/(I+1)\}_{I=1}^\infty\}$ is called $\delta$-continuous if there exists a constant $c_g$ such that for all $z_1, z_2 \in \mathcal{Z}$, there exists a $\delta^*$ such that if $\delta \leq \delta^*$, then $\|g_\delta(z_1) - g_\delta(z_2)\| \leq c_g\|z_1 - z_2\|$.*

It is worth mentioning that piece-wise constant approximation $g_\delta$ of Lipschitz function $g$ with partition size $\delta$ is $\delta$-continuous. Moreover, if $\lim_{\delta \to 0} g_\delta = g$, the $\delta$-continuity of $g_\delta$ is equivalent to the

Figure 4: Illustration of discrete nature of pixel observations for continuous motion. Considering a motion similar to that depicted in Figure 1, we assume that the object occupies a very small volume and its motion is recorded in continuous time periods. The left side illustrates the underlying motion of the object, while the middle side shows the evolution of pixel values during the recording process. The right side depicts the observed motion derived from these pixel values, reflecting the discrete nature of pixel observations.

Lipschitz continuity of $g$. We adopt this definition due to the discrete nature of pixel observations, where slight variations in $z$ might not be reflected in image, as will be illustrated in Section 3.2.

As discussed previously, to learn a continuous latent dynamical model, it is essential that the extracted latent states $Z = \mathcal{E}_\delta \circ \boldsymbol{I}_\delta \circ \mathcal{S}(z(t))$ evolve continuously over time. Therefore, our objectives for the autoencoder are as follows:

- Find the encoders $\{\mathcal{E}_\delta\}$ such that $\mathcal{E}_\delta \circ \boldsymbol{I}_\delta \circ \mathcal{S}(z)$ is $\delta$-continuous.
- Find the decoders $\{\mathcal{D}_\delta\}$ such that $\mathcal{D}_\delta \circ \mathcal{E}_\delta$ is approximately the identity mapping.

The second objective guarantees that the learned latent states are non-trivial, which is a standard requirement for autoencoders. The first objective ensures that the latent states evolve continuously with the underlying dynamics. If it can be achieved, with an additional assumption that the pixel size $\delta^*$ of the image is sufficiently small, then there exists a constant $c_\mathcal{E}$ such that

$$\|Z_n - Z_{n+1}\| = \|\mathcal{E}_{\delta^*} \circ \boldsymbol{I}_{\delta^*} \circ \mathcal{S}(z_n) - \mathcal{E}_{\delta^*} \circ \boldsymbol{I}_{\delta^*} \circ \mathcal{S}(z_{n+1})\| \le c_\mathcal{E} \|z_n - z_{n+1}\| \le c_\mathcal{E} M_f \Delta t.$$

This inequality implies that the resulting latent variables $\{Z_n^m\}_{n=0,\cdots,N,\ m=1,\cdots,M}$, corresponding to the input images (1), are discrete samplings of Lipschitz continuous trajectories. This property allows us to employ a continuous dynamical model for learning the latent dynamics and predicting future behavior by decoding the predicted latent states using the decoder. In the following sections, we analyze why standard CNN encoders may fail to achieve the first objective and how our proposed method overcomes this limitation.

## 3.2 MODELING OF MOTION

To illustrate the rationale behind the modified definition of $\delta$-continuity introduced in the previous section, we consider the example of rigid body motion in a two-dimensional plane. This type of motion, involving only translation and rotation, allows for a clear representation of the mapping $\mathcal{S}$, which is essential for the subsequent analysis.

The equation of rigid body motion on a two-dimensional plane can be expressed as follows

$$\dot{z} = f(z),\ z = (z^t, z^r),\ z^t = (\boldsymbol{r}_1, \cdots, \boldsymbol{r}_K),\ z^r = (\theta_1, \cdots, \theta_K), \tag{4}$$

where $K$ is the number of rigid bodies. The image corresponding to the state $z$ is given by

$$\mathcal{S}(z) = \cup_{k=1}^K \Phi_{\theta_k}(\Omega_k) + \boldsymbol{r}_k,\ \Omega_k \subset \mathbb{R}^2, \tag{5}$$

where $\boldsymbol{r}_k = (r_{k,1}, r_{k,2})$ and $\Phi_{\theta_k}$ represent the translation and rotation of the object, respectively. Details of this model can be found in Appendix A.1. Here we assume that $\Phi_{\theta_k}(\Omega_k) + \boldsymbol{r}_k \subset [0,1]^2$ for $k = 1, \cdots, K$ and that these sets are pairwise disjoint for all $z \in \mathcal{Z}$.

Suppose $K = 1$ and the motion only involves translation, we take $\Omega_1 = (0, w_1] \times (0, w_2]$ and further assume that $w_i/\delta$ is integer. In this case, we have:

$$\boldsymbol{I}_\delta(\mathcal{S}(z)) = \boldsymbol{I}_\delta(\mathcal{S}(\hat{z})),\ \text{where } \hat{z} = (\hat{r}_{1,1}, \hat{r}_{1,2}) \text{ and } \hat{r}_{1,i} = \max_{j \in \mathbb{Z}, j\delta \le r_{1,i}} j\delta.$$

This implies that variations smaller than $\delta$ may not be captured in the image, and thus, we can only track the dynamics of $\hat{z}$, the piece-wise constant approximation of $z$, as illustrated in Fig. 4. Consequently, the function $\mathcal{E}_\delta \circ \boldsymbol{I}_\delta \circ \mathcal{S}(z)$ cannot be continuous under standard definitions of continuity. Using our definition, we can readily verify that (1) $g_\delta(z) = \hat{z}$ is $\delta$-continuous; (2) $g_\delta(z) = \boldsymbol{I}_\delta \circ \mathcal{S}(z)$ is not $\delta$-continuous; and (3) $\hat{g} \circ g_\delta : \mathcal{Z} \to \mathbb{R}^d$ is $\delta$-continuous if $g_\delta : \mathcal{Z} \to \mathbb{R}^{\hat{d}}$ is $\delta$-continuous and $\hat{g} : \mathbb{R}^{\hat{d}} \to \mathbb{R}^d$ is Lipschitz continuous.

Figure 5: **(A)** Illustration of convolution operation. The red boxes represent the filter of size $\mathcal{O}(1)$ or $\mathcal{O}(1/\delta)$. The blue box represents the object. The solid line indicates its initial position, while the dashed line represents its position after motion. **(B)** The variation of latent states divided by $\Delta t$ for the two-body system, where the encoder is a one-layer CNN with parameters uniformly sampled from $[-1, 1]$. The shaded region represents one standard deviation. **(C)** Examples of motion where the positions of the objects after variation only partially overlap with their positions before variation.

## 3.3 WHY STANDARD CNN AUTOENCODERS FAIL

Since $\boldsymbol{I}_\delta \circ \mathcal{S}(z)$ is not $\delta$-continuous, the composition $\mathcal{E} \circ \boldsymbol{I}_\delta \circ \mathcal{S}(z)$ is generally not $\delta$-continuous if no further restrictions is imposed on $\mathcal{E}$. Next we illustrate this issue by an example of uniform motion.

Consider a vertical bar with a width of $\Delta$ undergoing uniform horizontal motion within the image, which is a two-dimensional extension of the motion depicted in Fig. 1. The equation governing this motion is $\dot{z} = 1$, $z(0) = 0$. We suppose $t \in [0, 1/2]$ to ensure that the bar remains within the image. Let $\mathcal{S}(z) = (z, z + \Delta) \times (0, 1]$ and assume $\Delta/\delta$ is an integer. The image of $z = n\delta$ is of the form

$$\boldsymbol{I}_\delta \circ \mathcal{S}(n\delta) = (\boldsymbol{0}_{(I+1)\times n}, \boldsymbol{1}_{(I+1)\times(\Delta/\delta+1)}, \boldsymbol{0}_{(I+1)\times(I-n-\Delta/\delta)}),$$

where $\boldsymbol{x}_{i_1 \times i_2}$ denotes the $i_1 \times i_2$ matrix whose elements are all $x$. We can verify that if $\boldsymbol{I}_0 = \boldsymbol{I}_\delta \circ \mathcal{S}(n\delta)$, then $[\boldsymbol{I}_1]_{0,0} = \sum_{j_1=n}^{\Delta/\delta} \sum_{j_2=0}^{J_1} [\boldsymbol{W}_1^\delta]_{j_1,j_2} := g_\delta(n\delta)$.

Suppose the step size for the motion is set to $2\Delta$. We consider the two cases:

1) the CNN filter size is of constant size i.e. $J_1 = \mathcal{O}(1)$, then we have $g_\delta(2\Delta) = 0$ and $g(0) = B_0 := \sum_{j_1=0}^{J_l} \sum_{j_2=0}^{J_1} \mathcal{W}_l(j_1\delta, j_2\delta)$ as $\delta \to 0$, where $B_0 \neq 0$ otherwise $\boldsymbol{I}_1$ is all zero. Then for any constant $c_{\boldsymbol{I}_1} > 0$, if $2\Delta \leq |B_0|/c_{\boldsymbol{I}_1}$, $|g_\delta(2\Delta) - g(0)| > c_{\boldsymbol{I}_1}|2\Delta|$. This indicates that $[\boldsymbol{I}_{0,0}]$ is not $\delta$-continuous, implying that the extracted latent states cannot be learned as a continuous dynamics.

2) the CNN filter size increases as image resolution increases, i.e. $J_1 = \mathcal{O}(1/\delta)$, then we have

$$g_\delta(2\Delta) - g_\delta(0) = \sum_{j_1=2\Delta/\delta}^{3\Delta/\delta} \sum_{j_2=0}^{J_1} \mathcal{W}_1(j_1\delta, j_2\delta)\varepsilon_1 - \sum_{j_1=0}^{\Delta/\delta} \sum_{j_2=0}^{J_1} \mathcal{W}_1(j_1\delta, j_2\delta)\varepsilon_1, \ \varepsilon_1 = \frac{\delta}{J_1\Delta}.$$

We assume $\mathcal{W}_1(j_1\delta, j_2\delta)$ are i.i.d. samples from a uniform distribution on $[-1, 1]$. Then both $g_\delta(2\Delta)$ and $g(0)$ are also independent samples from a uniform distribution on $[-1, 1]$. It follows that $|g_\delta(2\Delta) - g_\delta(0)| \leq c_g \|\Delta\|$ for a given $c_g$ with probability zero as $\Delta \to 0$. This implies that if no further restrictions is imposed on standard CNN filters, $[\boldsymbol{I}_{0,0}]$ is not $\delta$-continuous and the extracted latent states do not exhibit continuous dynamics with probability one.

This calculation is schematically illustrated in the left panel of Fig. 5. It is noted that in scenarios involving small-volume objects and limited overlap in their positions between steps, a standard CNN without additional constraints performs poorly in outputting continuous latent states. A numerical demonstration of this limitation for two-body systems is shown in the middle panel of Fig. 5. Several examples of motions where this failure may occur are shown on the right side of Fig. 5.

## 3.4 QUANTIFYING CONTINUITY OF CNN AUTOENCODERS

In the aforementioned counterexample, it is evident that a sufficient condition for ensuring the continuous evolution of latent states is that $\mathcal{W}_1$ is Lipschitz continuous. In this section, we will quantify the continuity of CNN autoencoders rigorously and extend this analysis to more general scenarios.

**Assumption 3.1.** *There exists a positive constant $M_\Delta$ and an integer $L^* < L$ such that, if $(i_1, i_2)/\lfloor I/\prod_{i=1}^l s_i \rfloor \notin [M_\Delta, 1 - M_\Delta]^2$, then $[\boldsymbol{I}_l]_{i_1,i_2} = 0$ for $l = 1, \cdots, L^* - 1$.*

This assumption holds in scenarios where the objects of interest are well-captured and located within the central region of the image. Alternatively, it can be satisfied by padding the input image with a sufficient number of zeros.

With these preliminaries, we next present the main theorem and provide its proof in Appendix A.2.

**Theorem 3.1.** *Assume that the underlying dynamical system is a rigid body motion (4) on a two-dimensional plane. If Assumption 3.1 hold, let $c_{\mathcal{W}}$ be constants satisfying*

$$\max_{l=1,\cdots,L^*} |\mathcal{W}_l(y_1) - \mathcal{W}_l(y_2)| \leq c_{\mathcal{W}} \|y_1 - y_2\|,$$

*and if $s_l = 2$ for $l = 1, \cdots, L^* - 1$, then for any $z_1 = (z_1^t, z_1^r), z_2 = (z_2^t, z_2^r) \in \mathcal{Z}$, we have*

$$\|\mathcal{E}_\delta \circ \boldsymbol{I}_\delta \circ \mathcal{S}(z_1) - \mathcal{E}_\delta \circ \boldsymbol{I}_\delta \circ \mathcal{S}(z_2)\| \leq C c_{\mathcal{W}} \|z_1^r - z_2^r\| + \frac{C c_{\mathcal{W}}}{2^{L^*-1}} \|z_1^t - z_2^t\|, \ \ as \ \delta \to 0.$$

*Here $C$ is a constant independent of $\delta$ and $z$.*

This theorem establishes a connection between the $\delta$-continuity of a CNN encoder and the continuity of its filters. To ensure that the latent states evolve continuously with the underlying dynamics, it is sufficient for the functions $\mathcal{W}_l$ representing the filters in the first few layers to be Lipschitz.

## 3.5 METHOD TO PRESERVE CONTINUITY OF CNN AUTOENCODERS

In this section, we discuss strategies to promote continuity of $\mathcal{W}_l$, thereby ensuring that functions $\mathcal{W}_l, \ l = 1, \cdots, L^*$ have a small constant $c_{\mathcal{W}}$ and $\mathcal{E}_\delta \circ \boldsymbol{I}_\delta \circ \mathcal{S}$ is $\delta$-continuous.

Note the fact that $\max_l \max_{j_1,j_2} |\mathcal{W}_l(j_1\delta, j_2\delta)|/(\lceil J_l/2 \rceil \delta) \leq c_{\mathcal{W}}$, larger filters are necessary to ensure continuity. It is worth mentioning that using a filter with the same size as the input image in CNN is essentially equivalent to using a fully connected neural network (FNN). While FNN encoders are commonly employed in baseline methods, they are often insufficient for second objective of reconstructing images with complex visual patterns, as demonstrated in Appendix A.3.4.

Focusing on images of size $3 \times 128 \times 256$, the parameters of the downsampling layers within the encoder are detailed in Table 1. The first three layers in this table use large filters (i.e., $L^* = 4$). The remaining five layers are standard convolution layers used for extracting latent features.

Table 1: Architecture of the encoder in CpAE

| Layer | 1 | 2 | 3 | 4 | 5 | 6 | 7 | 8 |
|---|---|---|---|---|---|---|---|---|
| Filter size | 12 | 12 | 12 | 4 | 4 | 4 | 4 | (3,4) |
| Stride | 2 | 2 | 2 | 2 | 2 | 2 | 2 | (1,2) |

As we only need the values of $\mathcal{W}_l$ on grids for computation, we recommend using the nonlocal operators method (Gilboa & Osher, 2007; 2009), an image processing technique that promotes image continuity. This approach requires only the following regularizer for the filters:

$$\mathcal{J} = \lambda_J \sum_{l=1}^{L^*} \sum_{i_1,i_2,j_1,j_2=-\hat{J}}^{J^l+\hat{J}} (W_{i_1,i_2}^l - W_{j_1,j_2}^l)^2 k\left((i_1\delta, i_2\delta), (j_1\delta, j_2\delta)\right), \tag{6}$$

where $\lambda_J$ is a weight hyperparameter, which is set to 1 by default, $k$ is a positive and symmetric function and the parameter $\hat{J} \geq 1$. Herein, we recommend employing the Gaussian kernel function $k(x) = e^{-\|x\|_2^2/\sigma^2}$ and setting $\hat{J} = 1$. And we apply the regularizer Eq. (6) to the filters of the first three layers to penalize large $c_{\mathcal{W}}$ and ensure the filters have appropriate continuity.

## 4 EXPERIMENTS

The benchmark methods used for comparison in this section include existing dynamical models such as Neural ODEs (Chen et al., 2018), Hamiltonian Neural Networks (HNNs) (Greydanus et al., 2019), and Symplectic Networks (SympNets) (Jin et al., 2020; 2023), coupled with standard autoencoders. Neural ODEs incorporate only the continuity prior. In contrast, HNNs and SympNets are structured dynamical models that leverage prior knowledge of Hamiltonian systems. With the goal of obtaining latent states that closely aligns with the assumed dynamics, these methods (Greydanus et al., 2019; Jin et al., 2023) typically train the latent dynamical models and the autoencoders simultaneously by minimizing the following loss function:

$$\mathcal{L} = \lambda \sum_{(x,y)\in\mathcal{T}} \left( \|\mathcal{D} \circ \mathcal{E}(x) - x\|_2^2 + \|\mathcal{D} \circ \mathcal{E}(y) - y\|_2^2 \right) + \sum_{(x,y)\in\mathcal{T}} \|\Phi \circ \mathcal{E}(x) - \mathcal{E}(y)\|_2^2,$$

where $\mathcal{T} = \left\{ (X_n^m, X_{n+1}^m) \right\}_{n=0,1,\cdots,N-1,\ m=1,\cdots,M}$ is the training dataset, $\Phi$ is the latent dynamical model detailed in Appendix A.3.1.

CpAEs are able to learn latent states that evolve continuously with time. Thus, we propose to learn the latent states and their corresponding latent dynamical models separately. ODEs are not only continuous over time but also preserve orientation, as characterized by the positive determinant of the Jacobian for the phase flow. Therefore, we employ VPNets (Zhu et al., 2022), which have the unit Jacobian determinant, for regularization. This regularization also helps penalize the large constant $C$ in Theorem 3.1. The loss function for CpAEs is defined as follows:

$$\mathcal{L}_{CpAE} = \sum\nolimits_{(x,y)\in\mathcal{T}} \|\mathcal{D} \circ \mathcal{E}(x) - x\|_2^2 + \mathcal{J}_R + \mathcal{J},$$

where $\mathcal{J}_R$ is given by $\mathcal{J}_R = \lambda_R \sum_{(x,y)\in\mathcal{T}} \|\Phi_{vp} \circ \mathcal{E}(x) - \mathcal{E}(y)\|_2^2 + \|\mathcal{D} \circ \Phi_{vp} \circ \mathcal{E}(x) - y\|_2^2$ and $\Phi_{vp}$ is a small VpNet, $\lambda_R$ is set to $1$ by default. After training the CpAEs, we separately learn a continuous dynamical model for the latent dynamics. In this section, the latent model for CpAEs is chosen to be a Neural ODE.

We also compare our proposed method with the hybrid scheme of neural state variable (Chen et al., 2022), a discrete model that has demonstrated impressive predictive accuracy. Note that the datasets used in our experiments, except for the first one, are obtained from Chen et al. (2022). Following their preprocessing steps, we concatenate two consecutive frames to form the data points. Parameters of all methods can be found in Appendix A.3.1.

Assessing the fidelity of learned latent dynamical models remains an open challenge. The quality of background reconstruction can significantly impact the Pixel Mean Squared Error (PMSE). In contrast, if there is already no overlap, large deviations in object position may not significantly increase PMSE. In this paper, we adopt a similar definition as in Jin et al. (2023); Botev et al. (2021) to compute the Valid Prediction Time (VPT) for evaluating a model's predictive ability:

$$\text{VPT} = \arg\max\nolimits_t \left\{ t \leq T \mid \text{PMSE}(X_\tau, \bar{X}_\tau) \leq \varepsilon,\ \forall \tau \leq t \right\},$$

where $\varepsilon$ is a threshold parameter, $X_\tau$ is the ground truth and $\bar{X}_\tau$ is the prediction at time $\tau$. Here we set $T = 1$ and $\varepsilon = 0.007$ for the first three datasets, and $\varepsilon = 0.0015$ for the last dataset. We found that once these thresholds are exceeded, there is a significant deviation in the predicted images. The VPT scores are averaged across all test trajectories. We also compute the Valid Prediction Frequency (VPF), which represents the frequency of test trajectories for which VPT $= 1$. The code accompanying the experiments is publicly available at `https://github.com/Aiqing-Zhu/CpAE`.

### 4.1 CONTINUITY OF LATENT STATES

We demonstrate the continuity using a simple circular motion dataset comprising 220 images ($48 \times 48$) captured every 0.1 seconds along a single trajectory, with 70 images for training and 150 for testing. A single hidden layer $48 \times 48$ CNN autoencoder with various regularizers is used to learn the latent states, followed by a Neural ODE to model their dynamics. After training, we show the latent states of the test images and the predicted dynamics. More detail are given in Appendix A.4.1.

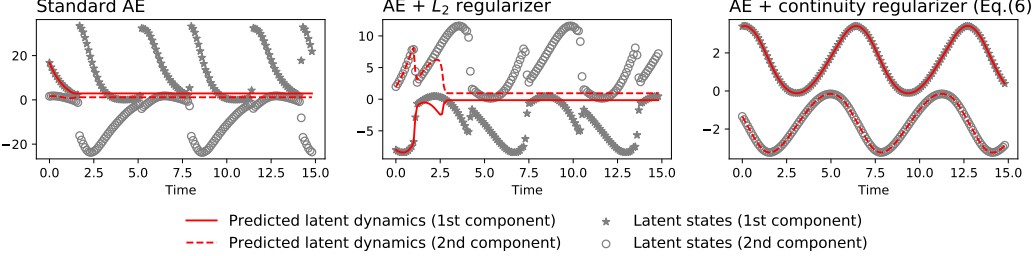

Figure 6: The latent states and the corresponding learned dynamics derived from them

As shown in Fig. 6, neither the standard autoencoder nor the addition of a conventional $L_2$ regularizer can extract continuously evolving latent states, leading to the failure of subsequent Neural ODE training. In contrast, the proposed continuity regularizer (6) ensures continuous latent state evolution, enabling the Neural ODE to effectively capture their dynamics.

| Method | CpAE | | Hybrid scheme | | AE +Neural ODE | | AE+HNN | | AE+SympNet | |
|---|---|---|---|---|---|---|---|---|---|---|
| Dataset | VPT | VPF | VPT | VPF | VPT | VPF | VPT | VPF | VPT | VPF |
| Damped pendulum | 99.2±8.5 | 99.2 | 95.4±15.0 | 88.3 | 50.7±31.2 | 23.3 | — | — | — | — |
| Elastic pendulum | 72.1±27.2 | 36.7 | 49.5±24.2 | 10.0 | 30.6±18.5 | 1.7 | — | — | — | — |
| Double pendulum | 69.1±31.5 | 40.0 | 46.8±21.4 | 4.6 | 24.3±13.8 | 0.0 | 11.0±4.2 | 0.0 | 15.1±12.8 | 0.0 |
| Swing stick | 57.4±20.4 | 11.1 | 13.7±5.1 | 0.0 | 14.4±7.5 | 0.0 | 24.1±14.5 | 0.0 | 14.8±12.2 | 0.0 |

Table 2: The performance of four physical systems evaluated using the VPT and VPF metrics. All values are scaled by a factor of $100$, with higher scores indicating better performance. VPT scores are reported as mean $\pm$ standard deviation. We do not report the performance of HNN and SympNet on the first two datasets, as their underlying systems are not Hamiltonian.

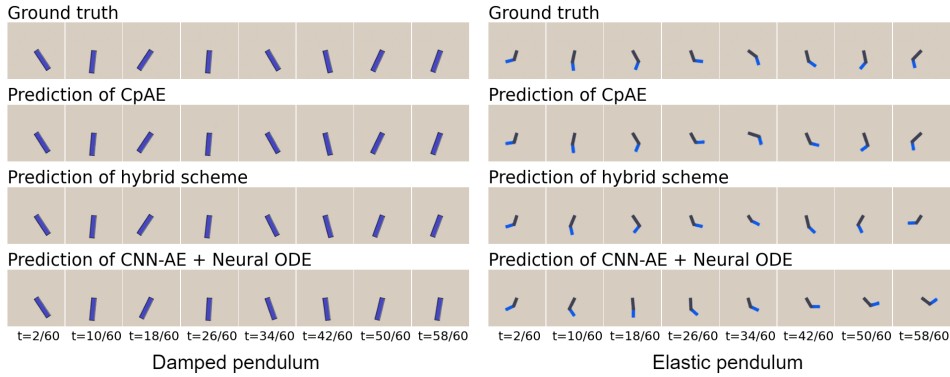

Figure 7: Predictions for simulation data.

## 4.2 SIMULATION DATA

We then benchmark on simulation datasets to show the enhanced prediction performance of CpAE.

**Damped pendulum**. This dataset consists of 1,200 trajectories, each containing 60 discrete data points-images of size $3 \times 128 \times 128$-sampled at a time interval of $\frac{1}{60}$ seconds. The details of this system are provided in Appendix A.3.2. The images of the physical state maintain the form of a single pendulum and are are generated using a similar procedure as in Chen et al. (2022).

**Elastic pendulum**. To verify the effectiveness of CpAEs on non-rigid motion, we consider the elastic double pendulum, where each pendulum arm can stretch and contract. The dataset is generated following the procedure outlined in Chen et al. (2022). It consists of 1,200 trajectories, each containing 60 data points-images of size $3 \times 128 \times 128$-sampled at time intervals of $\frac{1}{60}$ seconds.

Table 2 and Fig. 7 demonstrate that both the hybrid scheme with neural state variables and the proposed CpAEs accurately approximate the actual dynamics as they evolve. In contrast, standard CNN autoencoders exhibit lower predictive accuracy due to their tendency to produce discontinuous latent states (as illustrated in Appendix A.3.3), while using a continuous model to learn the latent dynamics. Since hybrid schemes do not require continuous latent variables, the advantage of CpAEs is less pronounced. Nevertheless, CpAEs provide a distinct benefit by yielding a continuous latent model, which is crucial for many scientific applications (Chen et al., 2023; Krishnapriyan et al., 2023; Qiu et al., 2022). This continuous latent model also facilitates tasks such as time reversal and interpolation between observed states (as shown in Appendix A.3.3).

## 4.3 REAL-WORLD DATA

To evaluate the model's performance on real-world systems, we perform experiments using the double pendulum and swing stick datasets from (Chen et al., 2022). Both datasets consist of images of size $3 \times 128 \times 128$ recorded at 60 fps. The double pendulum dataset contains $1,200$ trajectories, each consisting of 60 discrete image frames, while the swing stick dataset includes $85$ trajectories, each with $1,212$ discrete image frames.

Building on our success with simulated data, we further show that CpAEs outperform baseline methods on real-world data, as illustrated in Table 2 and Fig. 8. Although the double pendulum is a Hamiltonian system, and SympNets are well-suited for such systems (Jin et al., 2020), standard

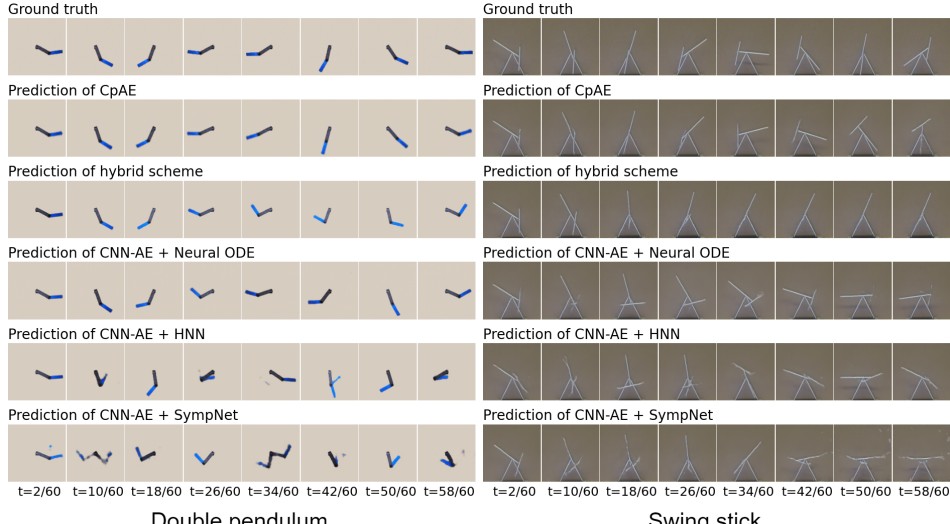

Figure 8: Predictions for real-world data.

autoencoders often struggle to learn latent variables that adhere strictly to Hamiltonian constraints, leading to predictions that deviate substantially from the true dynamics. This issue is particularly pronounced for the double pendulum system.

## 5 SUMMARY

Leveraging known properties of a data domain to design efficient deep learning architectures has significantly advanced the field. When using a dynamic model with prior knowledge to learn hidden dynamics from images, it is essential that the extracted latent states follow a dynamic system that aligns with the specified prior. In this paper, we introduce continuity-preserving autoencoders (CpAEs), a novel approach designed to impose the continuity restriction on latent state representations. Our main contribution is the mathematical formulation for learning dynamics from image frames, which illustrates the issue of discontinuity of the latent states extracted by standard CNN encoders. We then show that the output latent states of CNN encoders evolve continuously with the underlying dynamics, provided that the filters are Lipschitz continuous. Moreover, numerical experiments show that CpAEs outperform standard autoencoders.

In this paper, our continuity quantification is limited to rigid body motion in a two-dimensional plane. Generalizing our analysis to include non-rigid motion and the projection of three-dimensional motion onto a two-dimensional plane presents significant challenges, which we leave for future research. While CpAEs show superior performance compared to baseline approaches, it is important to note that they still fall short of effectively solving all tasks. As the visual complexity of the images increases, particularly in the challenging swing-stick tasks where the dynamics are not fully characterized, and the images are captured against complex backgrounds with significant noise, we observe a decline in performance, highlighting the need for more sophisticated approaches. Herein, we adopted the simplest method of promoting filter continuity by adding regularization. Future research could explore more effective methods, such as hypernetworks (Chauhan et al., 2023), to achieve this goal. Furthermore, our current research is exclusively focused on the weak prior of continuity within the context of CNN. Whereas numerous studies have explored incorporating classical mechanics-inspired inductive biases into neural networks to construct dynamical models, a promising direction would be to develop autoencoders that explicitly impose other forms of prior knowledge. Moreover, we would like to investigate the effectiveness of advanced architectures, such as Vision Transformers (Dosovitskiy et al., 2021) and their variants (Sriwastawa & Arul Jothi, 2024), as autoencoders for learning dynamics from images in future work. This approach holds promise because the operation applied to each patch in ViTs can be interpreted as a convolution with a kernel size equal to the patch size, and the mathematical formulation introduced is not restricted to CNNs.

ACKNOWLEDGMENTS

This research is supported by the National Research Foundation, Singapore under its AI Singapore Programme (AISG Award No: AISG3-RP-2022-028). A. Zhu is supported by the NRF fellowship (project No. NRF-NRFF13-2021-0005). Y. Pan is part of the programme DesCartes and is supported by the National Research Foundation, Prime Minister's Office, Singapore under its Campus for Research Excellence and Technological Enterprise (CREATE) programme.

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

# A APPENDIX

## A.1 RIGID MOTION MODELING

Here we consider rigid body motion, namely that all its particles maintain the same distance relative to each other. The position of the whole body can be represented by the imaginary translation and rotation that is needed to move the object from a reference placement to its current placement

We consider the motion of a rigid body, where all particles maintain a fixed distance relative to each other. The position of the entire body can be described by the combined translation and rotation required to move it from a reference placement to its current placement.

Let us first select a reference particle, denoted as $A$, typically chosen to coincide with the body's center of mass or centroid. When the rigid body is in its reference placement, the position vector of $A$ is denoted by $r_A^0$. For another particle $B$, its position vector can then be expressed as:

$$r_B^0 = r_A^0 + r_B^0 - r_A^0 = r_A^0 + |r_B^0 - r_A^0|(\cos\theta_B^0, \sin\theta_B^0)^\top,$$

where $|r_B^0 - r_A^0|$ denotes the distance between points $A$ and $B$, and $\theta_B^0$ represents the direction angle of the vector $r_B^0 - r_A^0$.

When the rigid body is in its current placement, we denote the position vector of $A$ by $r_A$. Then the position vector of particle $B$ can be written as

$$r_B = r_A + r_B - r_A = r_A + |r_B - r_A|(\cos\theta_B, \sin\theta_B)^\top.$$

Since the body is assumed to be rigid, the distance $|r_B - r_A| = |r_B^0 - r_A^0|$ remains invariant, and there exists an angle $\theta$ such that $\theta_B = \theta_B^0 + \theta$ for all particles $B$. Thus, we have:

$$r_B = r_A^0 + r_A - r_A^0 + |r_B^0 - r_A^0|(\cos\theta_B^0 + \theta, \sin\theta_B^0 + \theta)^\top$$
$$= r_A^0 + r_A - r_A^0 + |r_B^0 - r_A^0| \begin{pmatrix} \cos\theta & -\sin\theta \\ \sin\theta & \cos\theta \end{pmatrix} \begin{pmatrix} \cos\theta_B^0 \\ \sin\theta_B^0 \end{pmatrix}$$
$$= r_A - r_A^0 + \begin{pmatrix} \cos\theta & -\sin\theta \\ \sin\theta & \cos\theta \end{pmatrix} (r_B^0 - r_A^0) + r_A^0.$$

Thus, the position of the entire body can be described using the translation of the reference point $r = r_A - r_A^0$, and the orientation of the body $\theta$. Let $\Omega = \{r_B^0 \in \mathbb{R}^2 | B$ is the particle constitute the rigid body$\}$ denote the set of particle positions when the rigid body is in its reference placement. The set of positions corresponding to the state $z = (r, \theta)$ in the current placement can then be expressed as:

$$S(z) = r + \Phi_\theta(\Omega),$$

where $r = r_A - r_A^0$ represents the translation of the object, and $\Phi_\theta : \mathbb{R}^2 \to \mathbb{R}^2$ is defined as

$$\Phi_\theta(r_B^0) = \begin{pmatrix} \cos\theta & -\sin\theta \\ \sin\theta & \cos\theta \end{pmatrix} (r_B^0 - r_A^0) + r_A^0,$$

which represents the rotation of the object. Given that there are $K$ objects, the rigid body motion on a two-dimensional plane can be given by Eq. (4).

## A.2 PROOF OF THEOREM 3.1

### A.2.1 FUNCTION REPRESENTATION OF CNN AUTOENCODERS

For the convenience of analysis, we represent the feature maps and the filters of CNN autoencoder as functions defined on a two-dimensional plane:

$$\begin{aligned} &\text{Input:} && \mathcal{I}_0(y^0, z), \\ &\text{Hidden layers:} && \mathcal{I}_l(y, z) = \sum_{y^{l-1} \in Y^{l-1}} \mathcal{I}_{l-1}(y^{l-1}, z)\phi_l(y^{l-1}, y)\varepsilon_l, \ l = 1, \cdots, L, \\ &\text{Output:} && \mathcal{I}_L(y^L, z), \ y^L \in Y^L. \end{aligned} \quad (7)$$

where the input is defined as

$$\mathcal{I}_0(y, \ z) = \begin{cases} [\boldsymbol{I}_\delta \circ \mathcal{S}(z)]_{i_1, i_2}, & \text{if } y \in [i_1\delta, (i_1+1)\delta) \times [i_2\delta, (i_2+1)\delta), \quad i_1, i_2 = 0, \cdots, I, \\ 0, & \text{if } y \notin [0, 1]^2. \end{cases}$$

I.e., $\mathcal{I}_0$ is a piece-wise constant approximation of the indicator function of the set $\mathcal{S}(z)$, with a partition size of $\delta$. Here, we denote the evaluation set of the $l$-th layer as $Y^l \subset \mathbb{R}^2$, which is determined by the parameter stride:

$$Y^l = \{(i_1\delta_l, \ i_2\delta_l)\}_{i_1, i_2 = 0}^{\lfloor I/\prod_{i=1}^l s_i \rfloor + J_l}, \ \delta_l = (\prod_{i=1}^l s_i)\delta, \ l = 0, \cdots, L. \tag{8}$$

$\phi^l : \mathbb{R}^2 \times \mathbb{R}^2 \to \mathbb{R}$ represent the filter of $l$-th layer, i.e,

$$\phi_l\left(y^l + y, y^l\right) = \begin{cases} \mathcal{W}_l(j_1\delta, j_2\delta), & \text{if } y \in [j_1\delta_{l-1}, (j_1+1)\delta_{l-1}) \times [j_2\delta_{l-1}, (j_2+1)\delta_{l-1}), \\ & j_1, j_2 = 0, \cdots, J_l, \\ 0, & \text{otherwise.} \end{cases}$$

Then it is straightforward to verify that $\mathcal{I}_l : \mathbb{R}^2 \times \mathcal{Z} \to \mathbb{R}$ represents the function corresponding to the feature map $\boldsymbol{I}_l$, such that:

$$\mathcal{I}_l\big((i_1\delta_l, \ i_2\delta_l), \ z\big) = [\boldsymbol{I}_l]_{i_1, i_2}, \ l = 0, \cdots, L.$$

Refer to Fig. 9 for an illustration. All functions here depend on $\delta$. To avoid redundancy, we omit explicit notation of this dependence.

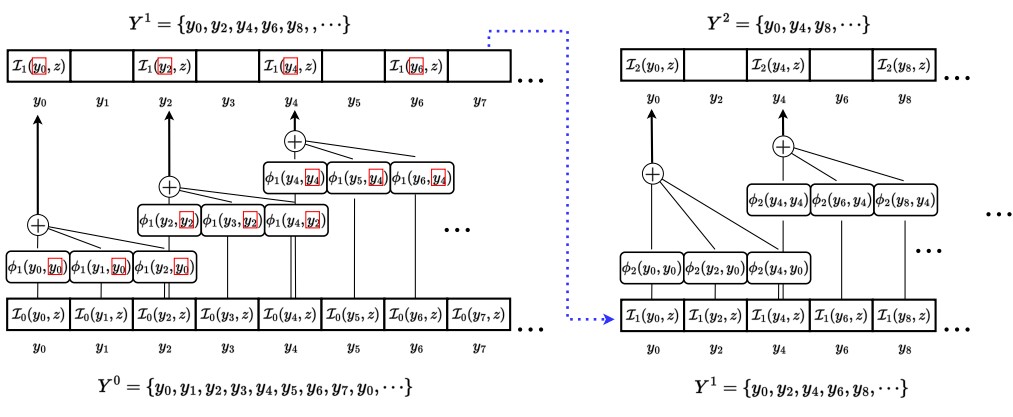

Figure 9: Illustration of the convolution operation. The stride of 2 halves the size of the feature map, and a new evaluation set is generated by removing half of the elements from the previous one, specifically by deleting every other element. The blue dashed lines indicate operations, such as activation layers, that do not change the evaluation set.

Under Eq. (7), Assumption 3.1 can be expressed in the following equivalent forms.

**Assumption A.1.** *There exist constant $M_\Delta$ such that if $\|\Delta\|_\infty \leq M_\Delta$, then*

$$\mathcal{I}_l(y^l, z) = 0, \ \forall y^l \in (Y^l \cup (Y^l - \Delta)) \backslash (Y^l \cap (Y^l - \Delta)).$$

Under Eq. (7), Theorem 3.1 has the following equivalent form.

**Theorem A.1.** *Assume that the underlying dynamical system is a rigid body motion (4) on a two-dimensional plane. If Assumption A.1, let $c_\phi$ be constant satisfying*

$$\max_{l=1, \cdots, L} \max_{y^l \in \mathbb{R}^2} |\phi_l(y_1^{l-1}, y^l) - \phi_l(y_2^{l-1}, y^l)| \leq \frac{c_\phi}{\prod_{i=1}^{l-1} s_i} \left\|y_1^{l-1} - y_2^{l-1}\right\|, \ \forall y_1^{l-1}, y_2^{l-1} \in \mathbb{R}^2,$$

*and if $s_l = 2$ for $l = 1, \cdots, L^* - 1$, then for any $z_1 = (z_1^t, z_1^r), z_2 = (z_2^t, z_2^r) \in \mathcal{Z}$,*

$$\|\mathcal{E} \circ \boldsymbol{I}_\delta \circ \mathcal{S}(z_1) - \mathcal{E} \circ \boldsymbol{I}_\delta \circ \mathcal{S}(z_2)\| \leq Cc_\phi \|z_1^r - z_2^r\| + \frac{Cc_\phi}{2^{L^*-1}} \left\|z_1^t - z_2^t\right\|, \ as \ \delta \to 0.$$

*Here $C$ is a constant independent of $\delta$ and $z$.*

Given that $\max_l \max_{j_1, j_2} |\phi_l(y_1^{l-1}, y^l)|/(\lceil J_l/2 \rceil \delta) \leq c_\phi$, we must choose $J_l = \mathcal{O}(1/\delta)$. Under this choice, the normalization coefficients $\varepsilon_l$ can be set as: $\varepsilon_1 = \delta^2/|\mathcal{S}(z)|$, $\varepsilon_l = \delta^2$ for $l = 2, \cdots, L$.

### A.2.2    THE CASE OF $L^* = 1$

We present the proof of Theorem A.1 for the case of $L^* = 1$.

Consider the general latent ODE (2). Assume that the image of $z$ can be represented as $\mathcal{S}(z)$. Furthermore, suppose there exist volume-preserving maps $\Phi_\Delta : \mathbb{R}^2 \to \mathbb{R}^2$, dependent on $\Delta \in \mathbb{R}^D$, such that $\Phi_\Delta(\mathcal{S}(z)) = \mathcal{S}(z + \Delta)$ and $\|\Phi_\Delta(x) - x\| \leq c_1 \|\Delta\|$ for some constant $c$. It is straightforward to verify that rigid body motions, including translation and rotation, satisfy these conditions.

Let

$$\tilde{\mathcal{I}}_1(y^1, z) = \frac{1}{|\mathcal{S}(z)|} \int_{\mathbb{R}^2} \mathcal{I}_0(y^0, z)\phi_1(y^0, y^1)dy^0 = \frac{1}{|\mathcal{S}(z)|} \int_{S(z)} \phi_1(y^0, y^1)dy^0,$$

we can verify that $\lim_{\delta \to 0} \mathcal{I}_1(y^1, z) = \tilde{\mathcal{I}}_1(y^1, z)$.

In addition, we have

$$|\tilde{\mathcal{I}}_1(y^1, z + \Delta) - \tilde{\mathcal{I}}_1(y^1, z)| = \frac{1}{|\mathcal{S}(z)|} \left| \int_{\mathcal{S}(z+\Delta)} \phi_1(y^0, y^1)dy^0 - \int_{\mathcal{S}(z)} \phi_1(y^0, y^1)dy^0 \right|$$

$$= \frac{1}{|\mathcal{S}(z)|} \left| \int_{\mathcal{S}(z)} \phi_1(\Phi_\Delta(y^0), y^1)dy^0 - \int_{\mathcal{S}(z)} \phi_1(y^0, y^1)dy^0 \right|$$

$$\leq \frac{1}{|\mathcal{S}(z)|} |\mathcal{S}(z)|c_1 c_\phi \|\Delta\| \leq c_1 c_\phi \|\Delta\|.$$

Taking $\delta^*$ such that for all $\delta \leq \delta^*$, the following inequalities hold:

$$|\mathcal{I}_1(y^1, z) - \tilde{\mathcal{I}}_1(y^1, z)| \leq c_1 c_\phi \|\Delta\|,$$
$$|\mathcal{I}_1(y^1, z + \Delta) - \tilde{\mathcal{I}}_1(y^1, z + \Delta)| \leq c_1 c_\phi \|\Delta\|.$$

These inequalities yield

$$|\mathcal{I}_1(y^1, z + \Delta) - \mathcal{I}_1(y^1, z)| \leq 3c_1 c_\phi \|\Delta\|,$$

which concludes the proof.

### A.2.3    THE CASE OF RIGID BODY TRANSLATIONAL MOTION

We now present the proof of Theorem A.1 for the case of rigid body motion involving only translation. We rewrite Eq. (7) in the following form to emphasize the dependence of evaluation sets.

Input:          $\mathcal{I}_0(y^0, z)$

Hidden layers:   $\mathcal{I}_{l+1}(y, z|Y^l, \cdots, Y^0) = \sum_{y^l \in Y^l} \mathcal{I}_l(y^l, z|Y^{l-1}, \cdots, Y^0)\phi_{l+1}(y^l, y),\ l \geq 0,$    (9)

Output:         $\mathcal{I}_L(y^L, z|Y^{L-1}, \cdots, Y^0),\ y^L \in Y^L.$

Under the above expression, if $Y^l$ is not of form given in Eq. (8), we can still compute the output.

**Definition A.1.** *Two evaluation sets $Y^l$ and $\hat{Y}^l$ are called equivalent, denoted as $Y^l = \hat{Y}^l$, if $\forall y^l \in Y^l \cup \hat{Y}^l \backslash Y^l \cap \hat{Y}^l$, we have $\mathcal{I}_l(y^l, z|Y^{l-1}, \cdots, Y^0) = 0$.*

We can readily check that two equivalent evaluation sets yield equivalent results for the convolution operation, i.e., if $Y^l = \hat{Y}^l$, $\mathcal{I}_{l+1}(y^{l+1}, z|Y^l, Y^{l-1}, \cdots, Y^0) = \mathcal{I}_{l+1}(y^{l+1}, z|\hat{Y}^l, Y^{l-1}, \cdots, Y^0)$.

Due to weight sharing, the convolution operation exhibits translational invariance. This property leads to the following characteristic of the function $\phi_l$ representing the weights of the filter.

**Property A.1.** *The function $\phi_l$ representing the weights of the convolution filter satisfies*

$$\phi_l(y^{l-1} + \Delta, y^l) = \phi_l(y^{l-1}, y^l - \Delta), \quad \forall \Delta \in \mathbb{R}^2.$$

For our motion model Eq. (4), if $K = 1$, the image of $z$ is translationally invariant due to Eq. (5), i.e., $\mathcal{I}_0(y, z + \Delta) = \mathcal{I}_0(y - \Delta, z)$. Similar property holds for each feature map.

**Lemma A.1.** *For motion model Eq. (4) with $K = 1$, the function $\mathcal{I}_l$ representing the feature map satisfies*

$$\mathcal{I}_l(y^l, z + \Delta | Y^{l-1}, \cdots, Y^0) = \mathcal{I}_l(y^l - \Delta, z | Y^{l-1} - \Delta, \cdots, Y^0 - \Delta), \quad \forall \Delta \in \mathbb{R}^2.$$

*Proof.* The case when $l = 0$ is obvious. Suppose now

$$\mathcal{I}_l(y^l, z + \Delta | Y^{l-1}, \cdots, Y^0) = \mathcal{I}_l(y^l - \Delta, z | Y^{l-1} - \Delta, \cdots, Y^0 - \Delta),$$

then,

$$
\begin{aligned}
&\mathcal{I}_{l+1}(y^{l+1}, z + \Delta | Y^l, \cdots, Y^0) \\
&= \sum_{y^l \in Y^l} \mathcal{I}_l(y^l, z + \Delta | Y^{l-1}, \cdots, Y^0) \phi_{l+1}(y^l, y^{l+1}) \varepsilon_l && \text{(by Eq. (9))} \\
&= \sum_{y^l \in Y^l} \mathcal{I}_l(y^l - \Delta, z | Y^{l-1} - \Delta, \cdots, Y^0 - \Delta) \phi_{l+1}(y^l, y^{l+1}) \varepsilon_l && \text{(by inductive hypothesis)} \\
&= \sum_{\hat{y}^l \in Y^l - \Delta} \mathcal{I}_l(\hat{y}^l, z | Y^{l-1} - \Delta, \cdots, Y^0 - \Delta) \phi_{l+1}(\hat{y}^l + \Delta, y^{l+1}) \varepsilon_l && \text{(by taking } \hat{y}^l = y^l - \Delta) \\
&= \sum_{\hat{y}^l \in Y^l - \Delta} \mathcal{I}_l(\hat{y}^l, z | Y^{l-1} - \Delta, \cdots, Y^0 - \Delta) \phi_{l+1}(\hat{y}^l, y^{l+1} - \Delta) \varepsilon_l && \text{(by Property A.1)} \\
&= \mathcal{I}_{l+1}(y^{l+1} - \Delta, z | Y^l - \Delta, \cdots, Y^0 - \Delta). && \text{(by Eq. (9))}
\end{aligned}
$$

Hence the induction holds and the proof is completed. $\square$

As a direct consequence, if $Y^{l-1} - \Delta = Y^{l-1}, \cdots, Y^0 - \Delta = Y^0$, we have the following corollary. Here, we omit the notation of evaluation set for brevity.

**Corollary A.1.** *For motion model Eq. (4) with $K = 1$, and further assume that $\Delta \in \mathbb{R}^2$ satisfies $Y^{l-1} - \Delta = Y^{l-1}$, we have*

$$\mathcal{I}_l(y^l, z + \Delta) = \mathcal{I}_l(y^l - \Delta, z).$$

**Corollary A.2.** *For motion model Eq. (4) with $K = 1$, and further assume that $\Delta$ satisfies $Y^{l-1} - \Delta = Y^{l-1}$, then we have*

$$\mathcal{I}_l(y^l, z + \Delta) - \mathcal{I}_l(y^l, z) = \sum_{y^{l-1} \in Y^{l-1}} \mathcal{I}_{l-1}(y^{l-1}, z) \left( \phi_l(y^{l-1} + \Delta, y^l) - \phi_l(y^{l-1}, y^l) \right) \varepsilon_l.$$

*Proof.*

$$
\begin{aligned}
&\mathcal{I}_l(y^l, z + \Delta) - \mathcal{I}_l(y^l, z) \\
&= \sum_{y^{l-1} \in Y^{l-1}} \left( \mathcal{I}_{l-1}(y^{l-1}, z + \Delta) - \mathcal{I}_{l-1}(y^{l-1}, z) \right) \phi_l(y^{l-1}, y^l) \varepsilon_l \\
&= \sum_{y^{l-1} \in Y^{l-1}} \mathcal{I}_{l-1}(y^{l-1} - \Delta, z) \phi_l(y^{l-1}, y^l) - \sum_{y^{l-1} \in Y^{l-1}} \mathcal{I}_{l-1}(y^{l-1}, z) \phi_l(y^{l-1}, y^l) \varepsilon_l \\
&= \sum_{\hat{y}^{l-1} \in Y^{l-1} - \Delta} \mathcal{I}_{l-1}(\hat{y}^{l-1}, z) \phi_l(\hat{y}^{l-1} + \Delta, y^l) - \sum_{y^{l-1} \in Y^{l-1}} \mathcal{I}_l(y^{l-1}, z) \phi_l(y^{l-1}, y^l) \varepsilon_l \\
&= \sum_{y^{l-1} \in Y^{l-1}} \mathcal{I}_{l-1}(y^{l-1}, z) \left( \phi_l(y^{l-1} + \Delta, y^l) - \phi_l(y^{l-1}, y^l) \right) \varepsilon_l.
\end{aligned}
$$

$\square$

For convenience, we introduce the following notations:

- For $\Delta$ satisfying $Y^0 - \Delta = Y^0$, we denote

$$a_l(\Delta) = \max_{y^l \in Y^l} |\mathcal{I}_l(y^l, z + \Delta) - \mathcal{I}_l(y^l, z)|,$$

- For $\Delta$ satisfying $Y^{l-1} - \Delta = Y^{l-1}$, we denote

$$b_l(\Delta) = \max_{y^l \in Y^l} \max_{y^{l-1} \in Y^{l-1}} |\phi_l(y^{l-1} + \Delta, y^l) - \phi_l(y^{l-1}, y^l)|$$

- We let $M_\mathcal{I}$ and $M_\phi$ be constants satisfying

$$M_\mathcal{I} = \max_l \max_{z \in \mathcal{Z}} \left( \sum_{y^l \in Y^l,\ y^l \neq 0} |\mathcal{I}_l(y^l, z)| \varepsilon_l \right), \quad M_\phi = \max_l \max_{y^l \in Y^l} \sum_{y^{l-1} \in Y^{l-1}} |\phi_l(y^{l-1}, y^l)| \varepsilon_l.$$

Then we have the following property.

**Lemma A.2.** *For motion model Eq. (4) with $K = 1$, let $\{\hat{\Delta}^l\}_{l=1}^L$ be a sequence defined recursively by $\hat{\Delta}^L = \Delta$ and for $l = L-1, \cdots, 1$, $\hat{\Delta}^l$ is a variation satisfying $Y^l - (\hat{\Delta}^{l+1} - \hat{\Delta}^l) = Y^l$. Then we have*

$$a_l(\hat{\Delta}^l) \leq M_\phi^I a_{l-I}(\hat{\Delta}^{l-I}) + \sum_{i=0}^{I-1} M_\phi^i M_\mathcal{I} b_{l-i}(\hat{\Delta}^{l-i} - \hat{\Delta}^{l-i-1}), \quad 1 \leq I \leq l \leq L.$$

*Proof.* Observing that $(\hat{\Delta}^l - \hat{\Delta}^{l-1})$ satisfies the condition in Corollary A.2, we have

$$|\mathcal{I}_l(y^l, z + \hat{\Delta}^l)) - \mathcal{I}_l(y^l, z)|$$
$$=|\mathcal{I}_l(y^l, z + \hat{\Delta}^{l-1} + (\hat{\Delta}^l - \hat{\Delta}^{l-1})) - \mathcal{I}_l(y^l, z + \hat{\Delta}^{l-1})| + |\mathcal{I}_l(y^l, z + \hat{\Delta}^{l-1}) - \mathcal{I}_l(y^l, z)|$$

$$\leq \left| \sum_{y^{l-1} \in Y^{l-1}} \mathcal{I}_{l-1}(y^{l-1}, z + \hat{\Delta}^{l-1}) \left( \phi_l(y^{l-1} + \hat{\Delta}^l - \hat{\Delta}^{l-1}, y^l) - \phi_l(y^{l-1}, y^l) \right) \varepsilon_l \right|$$

$$+ \left| \sum_{y^{l-1} \in Y^{l-1}} \left( \mathcal{I}_{l-1}(y^{l-1}, z + \hat{\Delta}^{l-1}) - \mathcal{I}_{l-1}(y^{l-1}, z) \right) \phi_l(y^{l-1}, y^l) \varepsilon_l \right|$$

$$\leq M_\mathcal{I} b_l(\hat{\Delta}^l - \hat{\Delta}^{l-1}) + M_\phi a_{l-1}(\hat{\Delta}^{l-1}).$$

This estimate can be rewritten as

$$a_l(\hat{\Delta}^l) \leq M_\mathcal{I} b_l(\hat{\Delta}^l - \hat{\Delta}^{l-1}) + M_\phi a_{l-1}(\hat{\Delta}^{l-1}),$$

which yields

$$a_l(\hat{\Delta}^l) \leq M_\phi^I a_{l-I}(\hat{\Delta}^{l-I}) + \sum_{i=0}^{I-1} M_\phi^i M_\mathcal{I} b_{l-i}(\hat{\Delta}^{l-i} - \hat{\Delta}^{l-i-1}),$$

and concludes the proof. $\square$

**Lemma A.3.** *For motion model Eq. (4) with $K \geq 1$, let $\Delta = (\Delta_1, \cdots, \Delta_K)$ with $\Delta_k \in \mathbb{R}^2$. let $\{\hat{\Delta}^l = (\hat{\Delta}_1^l, \cdots, \hat{\Delta}_K^l),\ \hat{\Delta}_k^l \in \mathbb{R}^2\}_{l=1}^L$ be a sequence defined recursively by $\hat{\Delta}^L = \Delta$ and for $l = L-1, \cdots, 1$, $\hat{\Delta}^l$ is a variation satisfying $Y^l - (\hat{\Delta}_k^{l+1} - \hat{\Delta}_k^l) = Y^l$ for $k = 1, \cdots, K$. If $\hat{\Delta}^0 = 0$, then we have*

$$a_l(\hat{\Delta}^l) \leq \sum_{k=1}^K \sum_{i=0}^{l-1} M_\phi^i M_\mathcal{I} b_{l-i}(\hat{\Delta}_k^{l-i} - \hat{\Delta}_k^{l-i-1}), \quad 1 \leq l \leq L.$$

*Proof.* Let $z = (\boldsymbol{r}_1, \cdots, \boldsymbol{r}_K) = (z_1, \cdots, z_K)$ and

$$\mathcal{I}_0(y^0, z) = \sum_{k=1}^K \mathcal{I}_0^k(y^l, z_k), \quad \mathcal{I}_l^k(y, z) = \sum_{y^{l-1} \in Y^{l-1}} \mathcal{I}_{l-1}^k(y^{l-1}, z_k) \phi_l(y^{l-1}, y) \varepsilon_l,\ l = 1, \cdots, L,$$

we next prove that $\mathcal{I}_l(y^l, z) = \sum_{k=1}^{K} \mathcal{I}_l^k(y^l, z_k)$ for $l = 1, \cdots, L$, by induction on $l$. Suppose now this statement holds for $l$, then we have

$$\mathcal{I}_{l+1}(y^{l+1}, z) = \sum_{y^l \in Y^l} \mathcal{I}_l(y^l, z)\phi_{l+1}(y^l, y^{l+1})\varepsilon_l = \sum_{y^l \in Y^l} \sum_{k=1}^{K} \mathcal{I}_l^k(y^l, z_k)\phi_{l+1}(y^l, y^{l+1})\varepsilon_l.$$

Swapping the order of summation leads to

$$\mathcal{I}_{l+1}(y^{l+1}, z) = \sum_{k=1}^{K} \sum_{y^l \in Y^l} \mathcal{I}_l^k(y^l, z_k)\phi_{l+1}(y^l, y^{l+1})\varepsilon_l = \sum_{k=1}^{K} \mathcal{I}_{l+1}^k(y^{l+1}, z_k),$$

which completes the induction. Applying Lemma A.2 to $\mathcal{I}_l^d$ we have

$$a_l(\hat{\Delta}^l) = \max_{y^l \in Y^l} |\mathcal{I}_l(y^l, z + \hat{\Delta}^l) - \mathcal{I}_l(y^l, z)| = \max_{y^l \in Y^l} |\sum_{k=1}^{D} \mathcal{I}_l^k(y^l, z_k + \hat{\Delta}_k^l) - \mathcal{I}_l^k(y^l, z)|$$

$$\leq \sum_{k=1}^{K} \max_{y^l \in Y^l} |\mathcal{I}_l^k(y^l, z_k + \hat{\Delta}_k^l) - \mathcal{I}_l^k(y^l, z)| \leq \sum_{k=1}^{K} \sum_{i=0}^{l-1} M_\phi^i M_\mathcal{I} b_{l-i}(\hat{\Delta}_k^{l-i} - \hat{\Delta}_k^{l-i-1}),$$

which concludes the proof. $\qquad\square$

By appropriately selecting the sequence $\{\hat{\Delta}^l\}_{l=1}^{L}$, we immediately have the following corollary.

**Corollary A.3.** *For motion model Eq. (4) with $K \geq 1$, suppose $\Delta = (\Delta_1, \cdots, \Delta_K)$ satisfies*

$$\Delta_k = (\Delta_{k,1}, \Delta_{k,2}) = (I_{k,1}\delta, \ I_{k,2}\delta) \in \mathbb{R}^2, \ \text{where } I_{k,1}, I_{k,2} \ \text{are integers.} \tag{10}$$

*Additionally, let $c_\phi$ be constants satisfying $b_l\left((i_1 \prod_{i=1}^{l-1} s_i)\delta, \ (i_2 \prod_{i=1}^{l-1} s_i)\delta\right) \leq c_\phi(|i_1|\delta + |i_2|\delta)$, and let $l_{k,j}^*$ is the largest integer satisfying $l_{k,j}^* \leq L$ and $2^{l_{k,j}^*-1} \leq |I_{k,j}|$ for $j = 1, 2$ and $k = 1, \cdots, K$. If we take $s_l = 2$ for $l = 1, \cdots, L - 1$, then we have*

$$a_L(\Delta) \leq \sum_{i=0}^{L-1} M_\phi^i M_\mathcal{I} \sum_{k=1}^{K} \left( \frac{c_\phi}{2^{l_{k,1}^*-1}} |\Delta_{k,1}| + \frac{c_\phi}{2^{l_{k,2}^*-1}} |\Delta_{k,2}| \right).$$

*Proof.* Let $\hat{\Delta}_k^l = \left(i_{k,1}^l \delta, \ i_{k,2}^l \delta\right) \in \mathbb{R}^2$, where $i_{k,1}^l, i_{k,2}^l$ are integers defined recursively by

$$i_{k,1}^L = I_{k,1}, \quad i_{k,1}^{l-1} = i_{k,1}^l \bmod 2^{l-1}, \quad l = L, \cdots, 2, \quad i_{k,1}^0 = 0;$$
$$i_{k,2}^L = I_{k,2}, \quad i_{k,2}^{l-1} = i_{k,2}^l \bmod 2^{l-1}, \quad l = L, \cdots, 2, \quad i_{k,2}^0 = 0.$$

We can readily check that the sequence $\{\hat{\Delta}^l = (\hat{\Delta}_1^l, \cdots, \hat{\Delta}_K^l)\}_{l=1}^{L}$ satisfies $Y^l - (\hat{\Delta}_k^{l+1} - \hat{\Delta}_k^l) = Y^l$ due to Assumption A.1 and $\hat{\Delta}^0 = 0$. In addition, we have

$$b_{l-i}(\hat{\Delta}_k^{l-i} - \hat{\Delta}_k^{l-i-1}) = b_{l-i}\left((i_{k,1}^{l-i} - i_{k,1}^{l-i-1})\delta, \ (i_{k,2}^{l-i} - i_{k,2}^{l-i-1})\delta\right)$$

$$= b_{l-i}\left(\left(\lfloor i_{k,1}^{l-i}/2^{l-i-1} \rfloor \cdot 2^{l-i-1}\right)\delta, \ \left(\lfloor i_{k,2}^{l-i}/2^{l-i-1} \rfloor \cdot 2^{l-i-1}\right)\delta\right)$$

$$\leq c_\phi \left| \lfloor i_{k,1}^{l-i}/2^{l-i-1} \rfloor \right| \delta + c_\phi \left| \lfloor i_{k,2}^{l-i}/2^{l-i-1} \rfloor \right| \delta$$

$$\leq \frac{c_\phi}{2^{l_{k,1}^*-1}} |i_{k,1}^L|\delta + \frac{c_\phi}{2^{l_{k,2}^*-1}} |i_{k,2}^L|\delta,$$

where $l_{k,j}^*$ is the largest integer satisfying $l_{k,j}^* \leq L$, $2^{l_{k,j}^*-1} \leq |i_{k,j}^L|$ for $j = 1, 2$ and $k = 1, \cdots, K$. Finally, applying Lemma A.3, we have

$$a_L(\Delta) \leq \sum_{i=0}^{L-1} M_\phi^i M_\mathcal{I} \sum_{k=1}^{K} \left( \frac{c_\phi}{2^{l_{k,1}^*-1}} |\Delta_{k,1}| + \frac{c_\phi}{2^{l_{k,2}^*-1}} |\Delta_{k,2}| \right).$$

$\qquad\square$

### A.2.4 PROOF OF THEOREM A.1

With these results, we are able to provided the proof of Theorem A.1.

*Proof of Theorem A.1.* Denote $\Delta = z_1 - z_2$, $\Delta^r = z_1^r - z_2^r$, $\Delta^t = z_1^t - z_2^t$ and for any $\delta$, denote

$$\hat{\Delta}^t = \arg \min_{\hat{\Delta} \text{ satisfies (10)}} \left\| \Delta^t - \hat{\Delta} \right\|, \quad \tilde{\Delta}^t = \Delta^t - \hat{\Delta}^t.$$

Then we have $\Delta = (\mathbf{0}, \Delta^r) + (\tilde{\Delta}^t, \mathbf{0}) + (\hat{\Delta}^t, \mathbf{0})$ and $\|\tilde{\Delta}^t\| \leq \delta$. Subsequently, we have

$$
\begin{aligned}
&\|\mathcal{E}_\delta \circ \boldsymbol{I}_\delta \circ \mathcal{S}(z_1) - \mathcal{E}_\delta \circ \boldsymbol{I}_\delta \circ \mathcal{S}(z_2)\| \\
\leq{}& \|\mathcal{E}_\delta \circ \boldsymbol{I}_\delta \circ \mathcal{S}(z_2 + (\mathbf{0}, \Delta^r)) - \mathcal{E}_\delta \circ \boldsymbol{I}_\delta \circ \mathcal{S}(z_2)\| \\
&+ \left\| \mathcal{E}_\delta \circ \boldsymbol{I}_\delta \circ \mathcal{S}(z_2 + (\mathbf{0}, \Delta^r) + (\mathbf{0}, \tilde{\Delta}^t)) - \mathcal{E}_\delta \circ \boldsymbol{I}_\delta \circ \mathcal{S}(z_2 + (\mathbf{0}, \Delta^r)) \right\| \\
&+ \left\| \mathcal{E}_\delta \circ \boldsymbol{I}_\delta \circ \mathcal{S}(z_2 + (\tilde{\Delta}^t, \Delta^r) + (\mathbf{0}, \hat{\Delta}^t)) - \mathcal{E}_\delta \circ \boldsymbol{I}_\delta \circ \mathcal{S}(z_2 + (\tilde{\Delta}^t, \Delta^r)) \right\|.
\end{aligned}
$$

By applying the conclusion for case of $L^* = 1$ to the first two components and Corollary A.3 to the third components, we obtain

$$
\begin{aligned}
&\|\mathcal{E}_\delta \circ \boldsymbol{I}_\delta \circ \mathcal{S}(z_1) - \mathcal{E}_\delta \circ \boldsymbol{I}_\delta \circ \mathcal{S}(z_2)\| \\
\leq{}& Cc_\phi \|z_1^r - z_2^r\| + Cc_\phi \delta + \sum_{i=0}^{L-1} M_\phi^i M_{\mathcal{I}} \sum_{k=1}^{K} \left( \frac{c_\phi}{2^{l_{k,1}^*-1}} |\Delta_{k,1}| + \frac{c_\phi}{2^{l_{k,2}^*-1}} |\Delta_{k,2}| \right).
\end{aligned}
$$

Let $\delta \to 0$, we conclude the proof. $\square$

### A.3 ADDITIONAL EXPERIMENTAL DETAILS

#### A.3.1 MODEL DETAILS

**Details of training**. The models are trained with Adam optimization for 500 epochs, using a learning rate of 0.001 and a batch size of 512. The data are divided into training, validation, and testing sets, with 80% used for training, 10% for validation, and 10% for testing.

**Dimensions of latent states**. For the two-body and damped pendulum systems, the latent state dimension was set to 2. For the other datasets, a latent state dimension of 8 was used. These values are intentionally larger than the theoretical minimum to enhance the model's expressiveness.

**CNN autoencoder**. The architecture of CNN encoder in this paper is adapted from the setting provided by Chen et al. (2022). The encoder is a 16-layer CNN, with the parameters of the downsampling convolutional layers detailed in Tables 3 and 4. Each downsampling convolutional layer is followed by an additional convolutional layer with same number of output channels, but a $3 \times 3$ filter, a stride of 1, and zero padding of 1 to enhance expressiveness. All convolutional layers are accompanied by a batch normalization layer and a ReLU activation function. The final latent variable is obtained through a linear layer.

Table 3: Architecture of the encoder in CpAE

| Layer | 1 | 2 | 3 | 4 | 5 | 6 | 7 | 8 |
|---|---|---|---|---|---|---|---|---|
| Filter size | 12 | 12 | 12 | 4 | 4 | 4 | 4 | (3,4) |
| Stride | 2 | 2 | 2 | 2 | 2 | 2 | 2 | (1,2) |
| #Filters | 16 | 32 | 64 | 64 | 64 | 64 | 64 | 64 |
| Padding | 5 | 5 | 5 | 1 | 1 | 1 | 1 | 1 |

Table 4: Architecture of the encoder in standard AE

| Layer | 1 | 2 | 3 | 4 | 5 | 6 | 7 | 8 |
|---|---|---|---|---|---|---|---|---|
| Filter size | 4 | 4 | 4 | 4 | 4 | 4 | 4 | (3,4) |
| Stride | 2 | 2 | 2 | 2 | 2 | 2 | 2 | (1,2) |
| #Filters | 16 | 32 | 64 | 64 | 64 | 64 | 64 | 64 |
| Padding | 1 | 1 | 1 | 1 | 1 | 1 | 1 | 1 |

All autoencoders share the same decoder architecture, which leverages residual connections and multi-scale predictions to to improve reconstruction performance. In addition to the deconvolutional layers specified in Table 5, each layer, except for the final one, is accompanied by an upsampling operation. This upsampling operation is implemented using a transposed convolutional layer with a filter size of $4\times4$, a stride of 2, and a Sigmoid activation function. The outputs of the deconvolutional layer and the upsampling layer are then concatenated along the channel dimension to form the input for the subsequent layer.

Table 5: Architecture of the decoder

| Layer | 8 | 7 | 6 | 5 | 4 | 3 | 2 | 1 |
|---|---|---|---|---|---|---|---|---|
| Filter size | (3,4) | 4 | 4 | 4 | 4 | 4 | 4 | 4 |
| Stride | (1,2) | 2 | 2 | 2 | 2 | 2 | 2 | 2 |
| #Filters | 64 | 64 | 64 | 64 | 64 | 32 | 16 | 3 |
| Padding | 1 | 1 | 1 | 1 | 1 | 1 | 1 | 1 |
| Activation | ReLU | ReLU | ReLU | ReLU | ReLU | ReLU | ReLU | Sigmoid |

**Filters size.** Larger filters are essential to ensure continuity. For a filter in the $l$-th layer of size $J_l \times J_l$, let $m_l = \max_{j_1,j_2} |\mathcal{W}_l(j_1\delta, j_2\delta)|$ represent the largest weight value. The distance from the location of this maximum value to the filter boundary must be less than $\lceil J_l/2 \rceil \delta$. This constraint leads to $\max_l m_l/(\lceil J_l/2 \rceil \delta) \leq c_{\mathcal{W}}$. Namely, $\max_l \max_{j_1,j_2} \frac{|\mathcal{W}_l(j_1\delta, j_2\delta)|}{\lceil J_l/2 \rceil \delta} \leq c_{\mathcal{W}}$. In Theorem 3.1, we show that the constant for the translation component is given by $\frac{c_{\mathcal{W}}}{2^{L^*-1}}$. Assuming $\frac{c_{\mathcal{W}}}{2^{L^*-1}}, \max_l \max_{j_1,j_2} |\mathcal{W}_l(j_1\delta, j_2\delta)| = \mathcal{O}(1)$, it follows that we only need to ensure $2^{L^*} J_l = \mathcal{O}(1/\delta)$. Based on this, we set $L^* = 4$ in our experiments. The images we use are of size $3 \times 128 \times 256$. This implies $2^4 J_l = 128$ or $2^4 J_l = 256$, which translates to $J_l = 8$ or $J_l = 16$. Taking the average of these values, we set $J_l = 12$.

**VPNet.** We employ a very small VPNet for regularization. It comprises three linear layers, each consisting of two sublayers. The activation function employed is Sigmoid.

**Neural ODE.** Neural Ordinary Differential Equations (Neural ODE) Chen et al. (2018) are continuous models by embedding neural networks into continuous dynamical systems. In this paper, we consider autonomous systems of first-order ordinary differential equations

$$\frac{d}{dt}y(t) = f_\theta(y(t)), \quad y(0) = x,$$

The governing function $f_\theta$ in the Neural ODE is parameterized by a FNN with two hidden layers, each containing 128 units and using a tanh activation function. Time discretization is performed using the explicit midpoint scheme.

**SympNet.** We utilize LA-SympNets with the default training configuration provided by (Jin et al., 2023). This setup includes three layers, each containing two sublayers, and employs the Sigmoid activation function.

**HNN.** The Hamiltonian in the HNN is parameterized by a FNN with two hidden layers, each containing 128 units and using a tanh activation function. Time discretization is performed using the explicit midpoint scheme.

### A.3.2 DETAILS OF DATASETS

**Damped single pendulum.** We consider a pendulum of length $L$ and mass $m$, attached to a fixed point by a rigid rod, free to swing under the influence of gravity. The moment of inertia of the pendulum is given by $I = \frac{1}{3}mL^2$. Let $\theta$ represent the angular position and $\dot{\theta}$ the angular velocity. The system's kinetic energy $T$ and potential energy $V$ are as follows:

$$T = \frac{1}{2}I\dot{\theta}^2 = \frac{1}{6}mL^2\dot{\theta}^2, \quad V = -\frac{1}{2}mgL\cos(\theta).$$

These energy yield the following second-order differential equation governing the system's dynamics:

$$\ddot{\theta} = -\frac{3g}{2L}\sin(\theta).$$

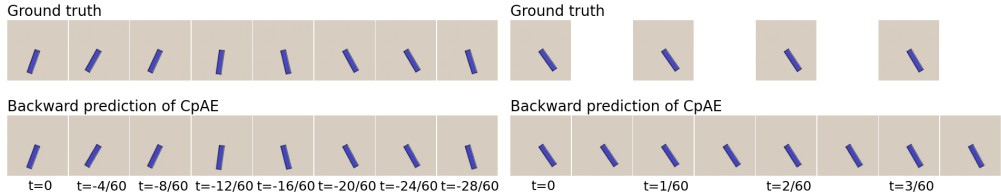

Figure 10: Backward predictions (left) and interpolations (right). Continuous models can accomplish these tasks by changing the value or the sign of $dt$ used in the integrator. This flexibility allows for both predicting past states and interpolating between known points.

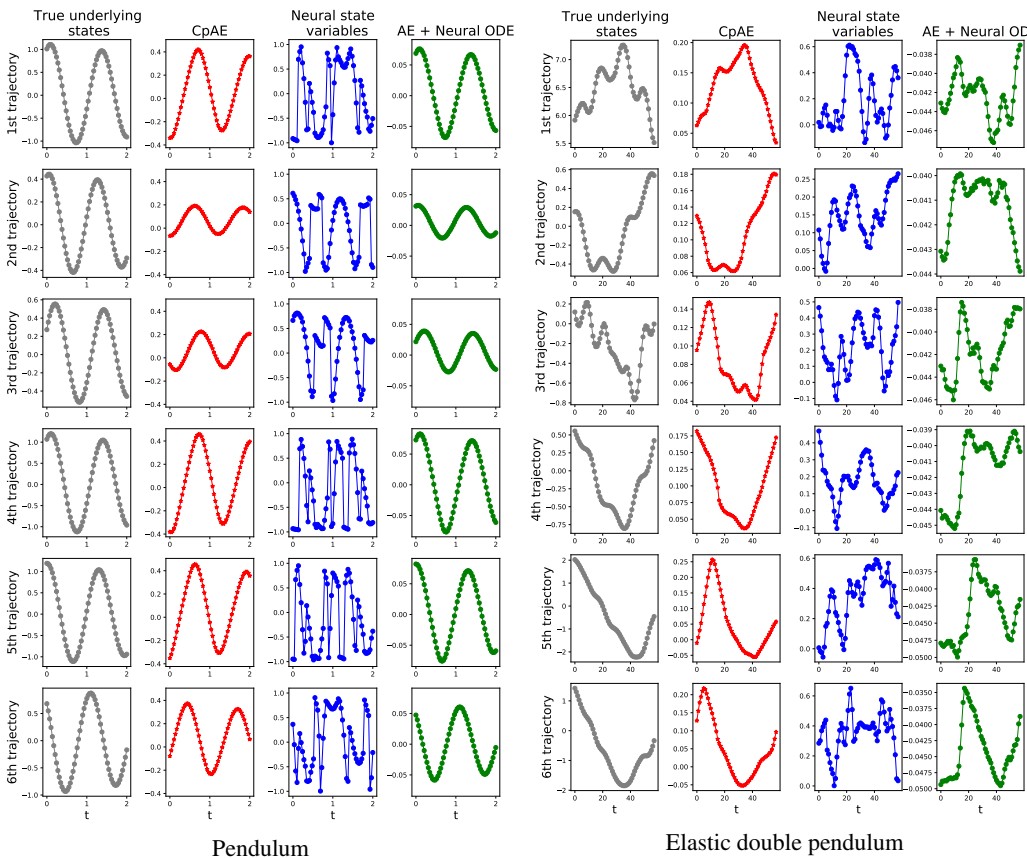

Figure 11: Evolution of the learned latent states for various autoencoders. CpAEs are capable of generating latent states that evolve continuously with time.

Here we account for a damping force proportional to the angular velocity, with a damping coefficient $k$, the final equation of motion for the damped pendulum becomes:

$$\ddot{\theta} = -\frac{3g}{2L}\sin(\theta) - k\dot{\theta}.$$

To simulate the physics equation of the single pendulum system, we set the pendulum mass $m = 1$, the pendulum length as $L = 0.125$ and the frictional constant $k = 0.8$. For each trajectory, we randomly sample the initial angular position $\theta$ and angular velocity $\dot{\theta}$.

### A.3.3 ADDITIONAL RESULTS

Here we show the continuous models allows for both predicting past states and interpolating between known points (Fig. 10), and we visualize the evolution of the learned latent states (Fig. 11).

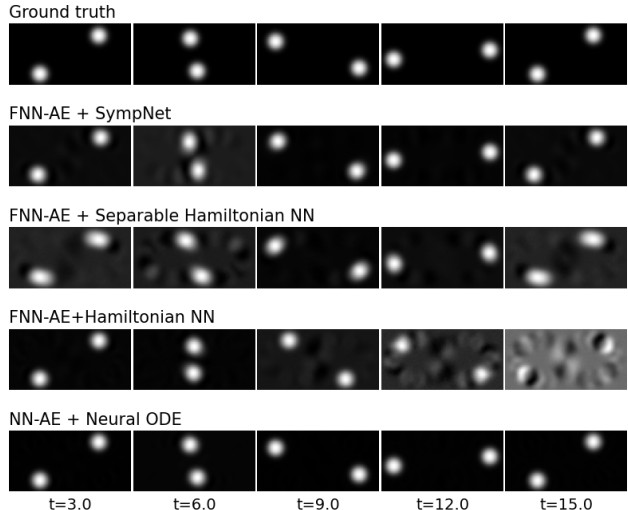

Figure 12: Predictions for two-body dataset.

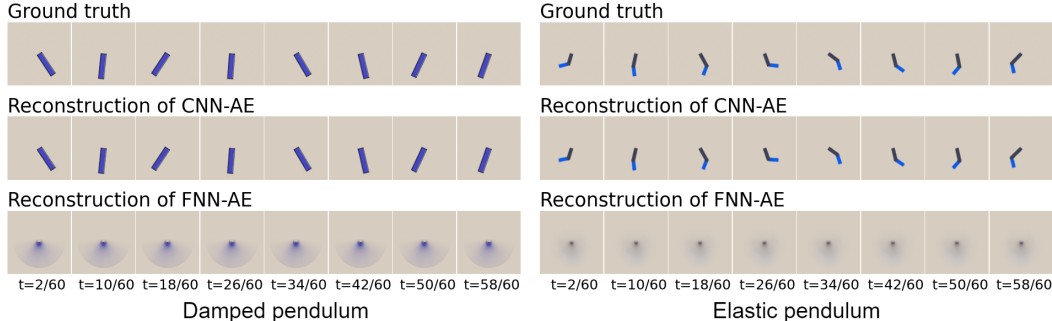

Figure 13: Reconstructions for simulation data.

### A.3.4 RECONSTRUCT IMAGE FROM EXACT LATENT STATES: FNN AUTOENCODER

We test the performance of FNN autoencoders using Two-body problem data. This dataset is generated in the same manner as in Jin et al. (2023) and is used to illustrate the FNN autoencoder. It consists of 100 observations-images of size $100 \times 50$-captured at a time interval of 0.6 seconds along a single trajectory.

As shown in Fig. 12, all FNN-based methods, except HNN, accurately predict the ground truth within the given time interval for the two-body system data. Given the strong performance baseline set by these established methods, any further improvements are likely to be incremental and may not significantly surpass the current results.

However, this success does not extend to datasets with more complex visual patterns. Fig. 13 illustrates that FNN autoencoders fail to achieve complete reconstruction for the damped pendulum and elastic double pendulum datasets. Here, reconstruction indicates that the encoder extracts the current latent state from the input image, and the decoder maps it back to reconstruct the same image. In this process, the decoder uses the exact latent state as input, with no dynamical information involved or reflected in these reconstruction results. As discussed in Section 2, the third goal of learning dynamics from images is to identify a decoder capable of reconstructing pixel-level observations using the exact latent state as input. Therefore, we focus on CNNs in this paper.

### A.4 ABLATION STUDIES

#### A.4.1 DETAILS FOR CONTINUITY OF LATENT STATES

The autoencoder model consists of a single-layer CNN with a $48 \times 48$ convolutional kernel and 32 output channels, followed by a linear mapping to generate the latent states. This simple task and

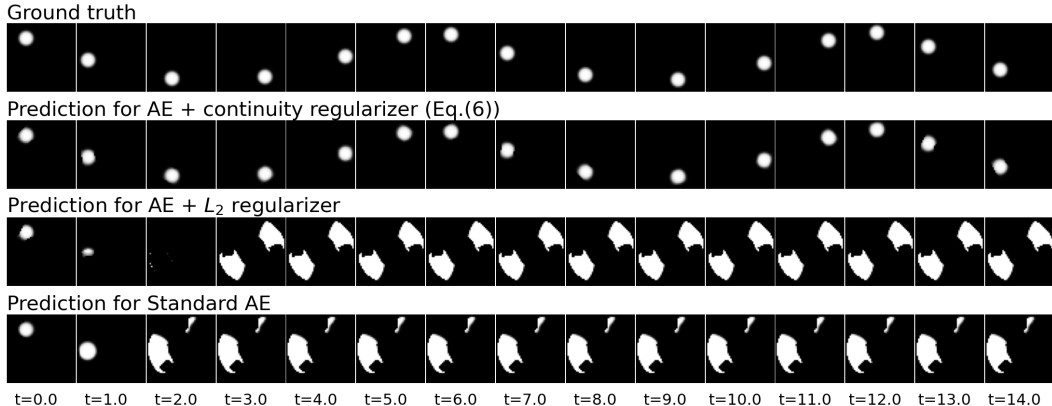

Figure 14: Predictions for circular motion data.

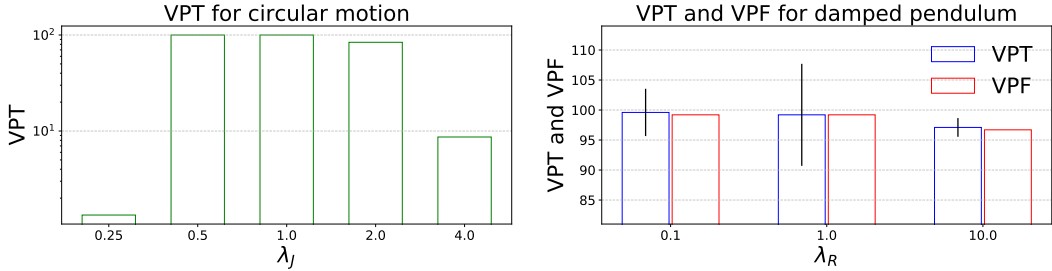

Figure 15: Results evaluated using the VPT and VPF metrics of varying weights. All values are scaled by a factor of 100, with higher scores indicating better performance. Note that the VPF metric is not reported for the circular motion dataset, as it consists of only a single trajectory.

model selection enable us to clearly and directly demonstrate the limitations of conventional CNNs in achieving $\delta$-continuity. Herein, all autoencoders, regardless of the applied regularizers, are trained using full-batch Adam optimization for $10^5$ epochs with a learning rate of 0.001. Once trained, a Neural ODE is employed to model the dynamics of the learned latent states. The governing function $f_\theta$ in the Neural ODE is parameterized by a FNN with single hidden layers containing 64 units and using a tanh activation function.

As a supplement to Fig. 6, the image prediction results are presented in Fig. 14. Since neither the standard autoencoder nor the addition of a conventional $L_2$ regularizer can predict the evolution of latent states, they fail to generate the predicted images. In contrast, the Neural ODE effectively captures the dynamics of the latent states generated by the proposed CpAE, enabling accurate predictions.

### A.4.2 EFFECTS OF WEIGHTS

Here we compare the performance when vary the weights of regularization terms $\lambda_J$ and $\lambda_R$.

The weight $\lambda_J$ controls the trade-off between model complexity and the continuity of the filter. As shown on the left side of Fig. 15, setting $\lambda_J$ too high penalizes overly complex models, which can lead to underfitting and suboptimal performance. Conversely, a lower $\lambda_J$ reduces the influence of regularization, increasing the risk of discontinuity in the learned latent states. The weight $\lambda_R$ emphasizes orientation preservation (i.e., a positive determinant of the Jacobian) in the learned latent states. As shown on the right side of Fig. 15, setting $\lambda_R$ too high shifts the focus toward volume preservation (i.e., a unit determinant of the Jacobian), which can cause underfitting and degrade model performance.

