# OpenReview forum: "Continuity-Preserving  Convolutional Autoencoders for Learning Continuous Latent Dynamical Models from Images"
_ICLR.cc/2025/Conference — ICLR 2025 Poster_

### Official Review · Reviewer_UqDE · 2024-10-22

**Soundness:** 3
**Presentation:** 3
**Contribution:** 3
**Rating:** 8
**Confidence:** 3

**Summary:**

This paper introduces continuity-preserving autoencoders (CpAEs) to learn continuous dynamical models and continuous latent states from the discrete pixel-based frames of a video.

First the authors present some theory showing that making the convolutional filters of the encoder Lipschitz continuous the states will evolve continuously. Then the authors introduce a regularizer to promote Lipschitz continuity.

The experiments show that by using CpAEs to learn continuous latent states, the learnt dynamical models are more accurate compared to standard CNN-based encoders that might produce discontinuous states .

**Strengths:**

1. Learning dynamical systems from image data is an important topic relevant in many applications.
2. Standard autoencoder are not forced to learn latent states that are continuous over time, which is a problem when trying to learn the underlying continuous model of the dynamics of the system. The paper presents an interesting theoretically grounded approach to enforce state continuity, which leads to improved performances in the considered experiments.
3. This theoretical framework could be useful for the community to build on.

**Weaknesses:**

1. My main concern on this paper is that I am not convinced that the proposed ideas will generalize well to more realistic tasks for which standard CNN encoders are powerful enough. While this is also noted by the authors as a limitation in section 5, I feel that for this paper to have a bigger impact on the ICLR community it would be important to apply this method to more complex architectures (e.g. VAEs, ViT) that will hopefully improve performances at least on the "swing stick" experiment.

2. The experimental section is missing some ablation studies to compare performances when varying some key parameters: 1) filter sizes and 2) regularizer parameters $\lambda_J$ and $\lambda_R$

3. The mathematical formulation proposed in section 3.1 to project states to pixels is key to understand the rest of the paper but not very easy to follow. It could benefit from using a simple running example, for example focusing on a single particle moving in space (if you add this example, you could even consider moving the example in section 3.2 to the appendix as a multi-particle extension)

4. Some notation is not introduced in the main text: $\Omega$ in section 3.1, $K$ in section 3.2, $\lambda_J$ in section 3.5, $\lambda$ and $\lambda_R$ in section 4

**Questions:**

I have two questions on your modelling choices. What do you mean when you say:
1. "Given our priority on ensuring the fidelity of the latent dynamical model, this work focuses on deterministic autoencoders."
2. "CpAEs are able to learn latent states that closely align with the assumed continuous dynamics. Thus, we propose to learn the latent states and their corresponding latent dynamical models separately"

---

> ### Author Response · Authors · 2024-11-21
> **Response to Reviewer UqDE (1/2)**
>
> - **Ideas are limited only to CNNs.**
>
>   Thank you for your valuable and constructive comments. We acknowledge that the analyses and algorithms presented in Sections 3.3, 3.4, and 3.5 are specifically tailored to CNN-based autoencoders (CNN-AEs). While we agree that extending our ideas to more complex architectures, such as Vision Transformers (ViTs), would be highly valuable, we also recognize that transformers have fundamentally distinct architectures and information processing mechanisms compared to CNNs. They may exhibit unique limitations regarding issues of discontinuity and thus would require tailored strategies to address.
>
>
>
>
>   However, the mathematical formulation introduced in Section 3.1, along with its explanation in Section 3.2, is not restricted to CNN architectures. Moreover, while the core operations differ significantly, other models, such as ViT, still share several underlying connections with CNNs. For instance, the operation applied to each patch in ViT is essentially a FNN, which can be interpreted as a CNN with a kernel size equal to the patch size. Based on your valuable feedback, we recognize the importance of extending our idea to more complex architectures like VAEs and ViTs. We would like to explore these directions further and apply our methods to more realistic and challenging tasks in a separate work.
>
>
>
>   In this paper, we focused on CNNs because they are widely recognized as a highly effective model for learning dynamics from images (e.g., Chen et al. (2022)). Their broad adoption and demonstrated success across diverse applications make them a compelling subject for analysis and enhancement, ensuring that our contributions are both practically relevant and readily applicable to real-world scenarios.
>
>
>   To address this concern, we have included a discussion in our revised manuscript to highlight the potential for extending our approach to other neural network architectures in future research in Section 5, page 10.
>
>
>   Boyuan Chen, Kuang Huang, Sunand Raghupathi, Ishaan Chandratreya, Qiang Du, and Hod Lipson. Automated discovery of fundamental variables hidden in experimental data. Nature Computational Science, 2(7):433–442, 2022.
>
> - **Lack of ablation studies.**
>
>   Thank you for your valuable comments. In the revised manuscript, we have included new numerical results dedicated to ablation studies to better understand the contribution of each component to the overall performance.
>
>   First, we have added a new experiment in Section 4.1 (page 8) of the revised manuscript to demonstrate that large filters alone cannot improve the continuity of latent states. And neither the standard autoencoder nor the addition of a conventional $L_2$ regularizer can extract continuously evolving latent states, leading to the failure of subsequent Neural ODE training. In contrast, the proposed continuity regularizer ensures continuous latent state evolution, enabling the Neural ODE to effectively capture their dynamics.
>
>   In addition, we compare the performance when varying the weights of regularization terms $\lambda_J$ and $\lambda_R$ in Appendix A.4.2, pages 23–24. The weight $\lambda_J$ controls the trade-off between model complexity and the continuity of the filter. As shown on the left side of Fig. 14, setting $\lambda_J$ too high penalizes overly complex models, which can lead to underfitting and suboptimal performance. Conversely, a lower level $\lambda_J$ reduces the influence of regularization, increasing the risk of discontinuity in the learned latent states. The weight $\lambda_R$ emphasizes orientation preservation (i.e., a positive determinant of the Jacobian) in the learned latent states. As shown on the right side of Fig. 14, setting $\lambda_R$ too high shifts the focus toward volume preservation (i.e., a unit determinant of the Jacobian), which can cause underfitting and degrade model performance.

---

> ### Author Response · Authors · 2024-11-21
> **Response to Reviewer UqDE (2/2)**
>
> - **Using a simple running example.**
>
>   We appreciate your suggestion to include a simple running example to improve clarity. In response to your feedback, we have added an example focusing on a single particle moving in space to section 3.1.  Additionally, we have moved the multi-particle example in section 3.2 to the appendix, as recommended.
>
> - **Some notation is not introduced.**
>
>   We are sorry for the lack of introduced notation in the main text. In the revised manuscript, we have added explicit introductions and explanations for any previously undefined notation. Specifically: $\Omega$ is a Borel set in $[0,1]^2$ representing the set of positions of particles constituting the objects; $K$ is the number of rigid bodies; $\lambda_J$ and $\lambda_R$ are weights of regularization and are set to 1.
>
> - **Question 1:**
>
>    By stating "Given our priority on ensuring the fidelity of the latent dynamical model, this work focuses on deterministic autoencoders", we mean that our primary objective is to ensure the continuous evolution of the latent states over time, which is connected to the continuity of the dynamical trajectory in time. This continuity of latent states over time is crucial for accurately learning the latent dynamical models within the latent space.
>
>    The key distinction between VAE and CpAE lies in the required continuity. VAEs are designed to ensure continuity of latent spaces. This continuity of latent spaces facilitates random sampling and interpolation, making them invaluable for generative modeling. However, incorporating stochasticity can compromise the temporal continuity of latent trajectories over time. Therefore, we focus on deterministic autoencoders.
>
>    In the revised manuscript, we have modified this sentence to: "Given our priority on ensuring the continuous evolution of the latent states, this work focuses on deterministic autoencoders." in page 3 to clearly state our focus.
>
> - **Question 2:**
>
>   When we state that "CpAEs are able to learn latent states that closely align with the assumed continuous dynamics. Thus, we propose to learn the latent states and their corresponding latent dynamical models separately", we mean that CpAEs can effectively capture latent representations that reflect the continuous latent dynamics. Consequently, our approach involves first learning the latent states and then using a dynamical model (e.g., Neural ODE) to independently and separately learn the corresponding latent dynamical models.
>
>   Standard autoencoders cannot achieve this target. Our ablation studies show that using a standard autoencoder to learn latent variables and then separately training the corresponding latent dynamical models is ineffective. Therefore, existing approaches such as HNN and SympNet, which are coupled with autoencoders, typically train the latent dynamical models and the autoencoders simultaneously. This simultaneous training approach allows the latent dynamical models to enforce constraints, regularizing the alignment of the latent states with the assumed dynamics.
>
>   In the revised manuscript, we have modified the sentence to: "CpAEs are able to learn latent states that evolve continuously with time. Thus, we propose to learn the latent states and their corresponding latent dynamical models separately." in page 8.
>
> Once again, we are grateful for the careful reading and valuable comments. We hope we have responded to all in an acceptable way and believe that the paper is quite improved because of this revision.

---

> > ### Comment · Reviewer_UqDE · 2024-11-23
> > **Increased score**
> >
> > Thank you for the clarifications and additional experiments. As a result I increased the score

---

### Official Review · Reviewer_NnyW · 2024-11-01

**Soundness:** 2
**Presentation:** 2
**Contribution:** 3
**Rating:** 6
**Confidence:** 2

**Summary:**

This paper proposes continuity-preserving autoencoders (CpAEs) for learning continuous latent states from a series of images that represent a continuous dynamical system. The proposed autoencoders resolve problems from a naive usage of Convolutional Neural Networks due to the district nature of images. The key contributions are a mathematical formulation for learning continuous dynamics from image data, the sufficient condition for the latent states will evolve continuously with the underlying dynamics, a regularizer to promote continuity of filters and empirical results.

**Strengths:**

- The paper presents an interesting mathematical formulation for learning continuous dynamical systems from images.
- The method presents strong empirical results that show the proposed method works.

**Weaknesses:**

- There is a lack of comparison with AEs that are not based on standard CNNs in the results.
- There are many assumptions that limit the formulation only to CNNs.
- There is a lack of ablation studies on the proposed methods parts.

**Questions:**

- How did you determine the filter size for Table 1?
- How much does the regularizer affect the results? Maybe the uplift in the result is just because of the filter sizes?
- In Figure 6, there is no image of CpAE on the two bodies. Why?
- Why are there no quantitative results for the two body data?

---

> ### Author Response · Authors · 2024-11-21
> **Response to Reviewer NnyW (1/2)**
>
> - **Assumptions limit the formulation only to CNNs**.
>
>   Thank you for your valuable feedback. We acknowledge that our analyses and algorithms in Sections 3.3, 3.4, and 3.5 are limited to CNN-AE. However, we chose to focus on CNNs because they are widely adopted for image processing and are an effective model for learning dynamics from images (e.g., Chen et al., (2022)). Their widespread use and success in various applications make them a significant subject for analysis and improvement, ensuring that our contributions have practical relevance and can be effectively applied in real-world scenarios.
>
>   To address this concern, we have revised the title, abstract, and introduction to better emphasize our focus on the CNN model.
>
>
>   However, the mathematical formulation in Section 3.1 and its explanation in Section 3.2 are not limited to CNN architectures. These formulations are also applicable to more complex architectures, such as Vision Transformers (ViT). Moreover, the operation applied to each patch in ViT is essentially a FNN, which can be interpreted as a CNN with a kernel size equal to the patch size. Based on your valuable feedback, we recognize the importance of extending our idea to more complex architectures like VAEs and ViTs. Given their distinct architectures, these models may exhibit different limitations and require unique approaches to address their challenges. We plan to explore these directions further and apply our methods to more realistic and challenging tasks in separate work. To address this concern, we have highlighted the potential for extending our approach to other neural network architectures in future research in Section 5, page 10.
>
>   Boyuan Chen, Kuang Huang, Sunand Raghupathi, Ishaan Chandratreya, Qiang Du, and Hod Lipson. Automated discovery of fundamental variables hidden in experimental data. Nature Computational Science, 2(7):433–442, 2022.
>
>
> - **Lack of comparison with AEs that are not based on standard CNNs**.
>
>   Thank you for the comments. We would like to clarify that our method is specifically designed for CNN-based autoencoders. While FNN-based autoencoders are another commonly used architecture, we have numerically demonstrated in our paper that they are effective primarily for relatively simple tasks, such as the two-body data. However, FNN-based autoencoders prove inadequate for handling datasets with complex visual patterns, such as the damped pendulum and elastic pendulum systems, as shown in Appendix A.3.4 in page 23 of the revised manuscript.
>
>   Given these observations, our work focuses on CNN-based autoencoders. Specifically, we analyze the limitations of standard CNN autoencoders in extracting latent states that evolve continuously over time and propose a novel approach to address this challenge effectively. And due to this reason, we solely demonstrate superior performance of our method over standard CNN autoencoders through numerical comparisons.
>
>
>   To ensure our focus is clear to readers and to prevent any potential misunderstandings, we have revised the manuscript to explicitly emphasize this point in the title, abstract, and introduction sections.
>
> - **Lack of ablation studies**.
>
>   Thank you for your valuable comments. In the revised manuscript, we have included new numerical results dedicated to ablation studies to better understand the contribution of each component to the overall performance.
>
>   First, we have added a new experiment in Section 4.1 (page 8) of the revised manuscript to demonstrate that large filters alone cannot improve the continuity of latent states. And neither the standard autoencoder nor the addition of a conventional $L_2$ regularizer can extract continuously evolving latent states, leading to the failure of subsequent Neural ODE training. In contrast, the proposed continuity regularizer ensures continuous latent state evolution, enabling the Neural ODE to effectively capture their dynamics.
>
>   In addition, we compare the performance when varying the weights of regularization terms $\lambda_J$ and $\lambda_R$ in Appendix A.4.2, pages 23–24. The weight $\lambda_J$ controls the trade-off between model complexity and the continuity of the filter. As shown on the left side of Fig. 14, setting $\lambda_J$ too high penalizes overly complex models, which can lead to underfitting and suboptimal performance. Conversely, a lower level $\lambda_J$ reduces the influence of regularization, increasing the risk of discontinuity in the learned latent states. The weight $\lambda_R$ emphasizes orientation preservation (i.e., a positive determinant of the Jacobian) in the learned latent states. As shown on the right side of Fig. 14, setting $\lambda_R$ too high shifts the focus toward volume preservation (i.e., a unit determinant of the Jacobian), which can cause underfitting and degrade model performance.

---

> ### Author Response · Authors · 2024-11-21
> **Response to Reviewer NnyW (2/2)**
>
> - **How did you determine the filter size for Table 1.**
>
>   Larger filters are essential to ensure continuity. For a filter in the $l$-th layer of size $J_l \times J_l$, let $m_l = \max_{j_1, j_2} |W_l(j_1\delta, j_2\delta)|$ represent the largest weight value. The distance from the location of this maximum value to the filter boundary must be less than $\lceil J_l/2 \rceil \delta$. This constraint leads to $\max_l m_l / (\lceil J_l/2 \rceil \delta) \leq c_{\mathcal{W}}$. Namely, $\max_l \max_{j_1, j_2} \frac{|W_l(j_1\delta, j_2\delta)|}{\lceil J_l/2 \rceil \delta} \leq c_{\mathcal{W}}$. In Theorem 3.1, we show that the constant for the translation component is given by $\frac{c_{\mathcal{W}}}{2^{L^*-1}}$. Assuming $\frac{c_{\mathcal{W}}}{2^{L^*-1}}, \max_l \max_{j_1, j_2} |W_l(j_1\delta, j_2\delta)| = \mathcal{O}(1)$, it follows that we only need to ensure $2^{L^*} J_l = \mathcal{O}(1/\delta)$.
>
>    Based on this, we set $L^* = 4$ in our experiments. The images we use are of size $3 \times 128 \times 256$, obtained by assembling two images as described in Chen et al. (2022). This implies $2^4 J_l = 128$ or $2^4 J_l = 256$, which translates to $J_l = 8$ or $J_l = 16$. Taking the average of these values, we set $J_l = 12$.
>
>    To address this concern, we have provided a detailed explanation of setting in Appendix 4.3.1, page 21 of the revised manuscript.
>
>
> - **How much does the regularizer affect the results?**
>
>   Thanks for the helpful comments. We have included a new experiment in the revised manuscript (Section 4.1, page 8), where we apply CNN autoencoder with large filter and various regularizers to learn the latent states
>
>   As shown in Figure 6 in the revised manuscript, neither the standard autoencoder nor the addition of a conventional $L_2$ regularizer can extract continuously evolving latent states, leading to the failure of subsequent Neural ODE training. In contrast, the proposed continuity regularizer ensures continuous latent state evolution, enabling the Neural ODE to effectively capture their dynamics.
>
>
>
> - **In Figure 6 (now Figure 7), there is no image of CpAE on the two bodies**.
>
>   We would like to clarify that our intention in including the results of the two-body data was not to claim that our method outperforms FNN-AE, but rather to explain our motivation for focusing on CNNs.
>
>   FNN-AE is a commonly used architecture for learning dynamics from images. Given its prevalence, we have numerically demonstrated in our paper that FNN-AE is effective primarily for relatively simple tasks, such as the two-body data discussed in Section 4. Given the strong performance baseline set by these established methods, any further improvements are likely to be incremental and may not significantly surpass the current results. Thus, we did not apply more complex CNNs to this task. However, FNN-based autoencoders prove inadequate for handling datasets with complex visual patterns, such as the damped pendulum and elastic pendulum systems also examined in Section 4. Given these observations, our work focuses on CNN-based autoencoders.
>
>   Based on your feedback, we realize that including the results of the two-body data in the main text could cause unnecessary confusion. In response, we have moved the FNN experiments to Appendix A.3.4, page 23, in our revised manuscript. We have also highlighted that while FNN is highly effective for relatively simple tasks, it performs poorly on more complex ones.
>
>
>
>
>
> - **Why are there no quantitative results for the two body data**.
>
>   Due to the reasons mentioned above—namely, that FNN is effective for the two-body data—further improvements are likely to be incremental and may not significantly surpass the current results. Therefore, we did not apply our method or other baseline models based on CNNs to this dataset. As a result, there are no quantitative results for these models on the two-body data.
>
>
> Once again, we are grateful for the valuable comments. We hope we have responded to this concern in an acceptable way and believe that the paper is quite improved because of this revision.

---

> ### Comment · Reviewer_NnyW · 2024-11-25
> **Thank for the response**
>
> Thank you for your detailed and thoughtful response. Based on your clarifications and updates, I will raise my score.

---

### Official Review · Reviewer_UE7U · 2024-11-04

**Soundness:** 2
**Presentation:** 4
**Contribution:** 3
**Rating:** 6
**Confidence:** 4

**Summary:**

The authors consider the setting where observations of a continuous dynamical system are captured by a sequence of images which are discrete in time and space. The challenge they claim to address is the misalignment of the evolution of images in pixel coordinates and the evolution of the underlying continuous time dynamical system. To address this challenge, the authors introduce the notion of delta-continuous, a relaxation of Lipschitz continuity where the ratio of the change in latent states and the change in continuous dynamical states is bounded by a constant. The term delta in the name delta-continuous refers to the size of the image pixels. As I understand, the claim is that if the pixels are too big then the delta-continuous condition fails and the latent states cannot be learned as a continuous dynamics. The authors subsequently introduce the continuity-preserving autoencoders (CpAEs) to encourage latent states to evolve continuously with the underlying dynamics. CpAEs do this using large filters in the first few layers, and a regularizer on the weights that encourage image continuity.

**Strengths:**

The paper is well organized and well written, although in some parts it might be easier to read if variable names were augmented with the concept they represent.

**Weaknesses:**

While the ideas presented are interesting, the impact of conventional CNNs failing to be delta-continuous is not clear to me. The experiments in the main paper (e.g., Figure 6) attempt to demonstrate the importance of delta-continuity by comparing the reconstructive ability of CpAE to other networks. I am unconvinced by this experiment for the following reasons: 1) it is not clear to me that reconstructive ability is an appropriate indicator of latent continuity; 2) it is not clear to me that using a fully-connected architecture as a baseline is appropriate since ‘[FNNs] are often insufficient for … reconstructing images with complex visual patterns’ (Line 346); 3) the conventional CNN baseline has comparable reconstructive ability to the proposed model. In my view, a more appropriate experiment is presented in Figure 10, pairing this experiment with a quantitative assessment would be more convincing (also I’m not sure what ‘Neural state variables’ are). This alone however, would only demonstrate that the resulting latent structure is more smooth with the CpAE structure. An additional experiment showing that a fixed dynamical model is better able to model the dynamics of CpAE states than AE states would help.

**Questions:**

I am unclear about the following experimental choices: 1) why do the authors compare to HNN architectures if the goal of the paper is to showcase the impact of the CpAE structure? 2) how does removing the VPNet regularizer impact CpAE performance? 3) how does using smaller weights in the initial layers impact CpAE performance? 4) if the CpAE ensures latent continuity why use a latent similarity loss (Line 399)? How much does this loss contribute to the observed performance?

---

> ### Author Response · Authors · 2024-11-21
> **Response to Reviewer UE7U (1/3)**
>
> Thank you for your insightful feedback on our manuscript. Please find below a detailed response.
>
> - **The conventional CNN baseline has comparable reconstructive ability**.
>
>   Thank you for your valuable comments. We are sorry for not clearly distinguishing between prediction and reconstruction in the original manuscript. We have explained this point in detail in the revised version.
>
>
>
>    The **reconstruction results** in Figure 6 in the original paper are just the output of the decoder without any prediction on the latent dynamics. Specifically, the encoder extracts the current latent state from the input image, and the decoder maps it back to reconstruct the same image. In this process, the decoder uses the exact latent state as input, and no dynamical information is involved or reflected in these reconstruction results. The purpose of showing these reconstruction results is to numerically highlight that an FNN autoencoder struggles to accurately reconstruct complex images. We acknowledge that presenting the reconstruction and prediction results in a single graph may have been misleading, so we have updated the manuscript accordingly and moved the reconstruction results to Appendix A.3.4, page 23, in our revised manuscript.
>
>
>    In contrast, the **prediction results** in Figure 6  aim to evaluate model performance and involve the prediction on the latent space. The encoder first infers the initial latent state; the dynamical model then recursively predicts future latent states; and the decoder generates images by decoding these predicted latent states. In this process, the decoder can only generate accurate images if the dynamic model accurately predicts the accurate latent states. As demonstrated in Table 2, Figure 6, and Figure 7, the proposed CpAEs achieve superior prediction performance and outperform the baseline methods.
>
>
> - **Using reconstructive ability as an indicator of latent continuity**.
>
>    Thank you for your valuable technical comments. We would like to clarify that the accuracy of the predicted image serves as an indicator of the fidelity of the latent dynamical model.
>
>    Since the data comprises only image observations and the model is trained directly on these observations, the accuracy of the predicted images—decoded from the latent states predicted by the latent dynamical model—is the most commonly used error metric. The reconstruction results presented in the original paper demonstrate that the CNN decoder can accurately reconstruct the image if the latent state is precise. Therefore, in our task, the accuracy of the predicted images provides a reliable measure of the accuracy of the latent dynamical model.
>
>
>   To demonstrate the continuity of latent states and highlight the impact of conventional CNN autoencoders failing to be delta-continuous, we have included a new experiment based on your suggestion in the revised manuscript (Section 4.1, page 8).
>
>
>   Standard autoencoder latent variables often capture only statistical information, lacking the dynamic properties necessary for effective modeling. Consequently, applying a dynamical model to such latent variables cannot produce meaningful results. As shown in Figure 6 in the revised manuscript, neither the standard autoencoder nor the addition of a conventional $L_2$ regularizer can extract continuously evolving latent states, leading to the failure of subsequent Neural ODE training. In contrast, the proposed continuity regularizer ensures continuous latent state evolution, enabling the Neural ODE to effectively capture their dynamics.

---

> ### Author Response · Authors · 2024-11-21
> **Response to Reviewer UE7U (2/3)**
>
> - **Impact of conventional CNNs failing to be delta-continuous**.
>
>   As discussed in Section 2, the objectives of learning dynamics from images are:
>   1. To learn an encoder that extracts latent states consistent with the assumed latent dynamical system;  in particular, the extracted latent states have to be continuous in time.
>   2. To develop a dynamical model that accurately captures the underlying latent dynamics; and
>   3. To build a decoder capable of reconstructing the pixel observations.
>
>    The first objective is fundamental to the entire task; without it, no target dynamical system with the assumed structures can be identified. Specifically, learning a continuous latent dynamical model requires that the extracted latent states are discrete samples of Lipschitz continuous trajectories. Without this, directly fitting a continuous dynamical model to discontinuous latent states cannot yield a meaningful and valuable dynamical model.
>
>     The reconstruction results in original Figure 6 (now Figure 13 on page 23) demonstrate that the CNN-AE successfully achieves the third objective. Furthermore, numerous studies have shown that dynamical models can effectively capture system dynamics from data sampled from the target dynamical system (see, e.g., Chen et al., 2018; González-García et al., 1998; Raissi et al., 2018; Wang et al., 2021; Wu & Xiu, 2020; Xie et al., 2024, as cited in our manuscript). Based on this, we conclude that the performance drop in existing learning models is primarily due to the inability of standard autoencoders to extract latent states consistent with the assumed latent dynamical system.
>
>
>     As demonstrated in our experiment (Section 4.1, page 8) of the revised manuscript, the standard autoencoder is not able to extract continuously evolving latent states, leading to the failure of subsequent Neural ODE training.
>
>
>
> - **Augment variable names to reflect the concepts they represent**.
>
>    Thanks for the helpful comments. In the revised manuscript, we have added explicit introductions and explanations for any previously undefined variables. Specifically: In Section 3.1, page 4, $\Omega$ is a Borel set in $[0,1]^2$ representing the set of positions of particles constituting the objects; in Section 3.2, page 5, $K$ is the number of rigid bodies; in Sections 3.5 and 4, pages 7 and 8, $\lambda_J$ and $\lambda_R$ are weights of regularization and are set to 1.
>
> - **Using FNN as a baseline**.
>
>    Our intention in including the results of the FNN model was not to claim that our method outperforms FNN, but rather to numerically highlight that FNN, which is commonly used in existing methods, is inadequate for reconstructing images with complex visual patterns.
>
>
>    Based on your valuable feedback, we have moved the FNN experiments to Appendix A.3.4, page 23 in the revised manuscript to avoid any unnecessary confusion. We have also highlighted that while FNN is highly effective for relatively simple tasks, it performs poorly on more complex ones.
>
> - **Why compare to HNN**.
>
>    HNNs are latent dynamical models that can be coupled with any autoencoder. Therefore, we compare our method with HNNs coupled with a standard autoencoder.
>
>    As you suggested, using a dynamical model to learn the latent states of a standard autoencoder is indeed a reasonable baseline. However, this approach performs poorly in practice. For this reason, we included better-established and widely recognized methods for learning dynamics from images—such as HNN+AE, SympNet+AE, and NeuralODE+AE—as baselines. These methods couple latent dynamical models with autoencoders and train them simultaneously. By doing so, the latent dynamical models enforce constraints during training to regularize the alignment of the latent states with the assumed dynamics.
>
>
>    To address your concern, we have highlighted this point in Section 4, page 7, of the revised manuscript.

---

> ### Author Response · Authors · 2024-11-21
> **Response to Reviewer UE7U (3/3)**
>
> - **Effect of VPNet regularizer**.
>
>   In the newly included experiment based on your suggestion (Section 4.1, page 8), we employed only a large filter and the proposed continuity regularizer without incorporating VPNet.
>   As shown in Figure 6 in the revised manuscript, the proposed continuity regularizer ensures continuous latent state evolution. This simple task and model selection enable us to clearly and directly demonstrate the effect of the employed regularization.
>
>   Continuity over time is a necessary condition of the trajectories of ODEs. Additionally, ODE trajectories preserve orientation, as characterized by the positive determinant of the Jacobian for the phase flow. Our use of VPNet, which has a unit Jacobian determinant, is specifically intended to regularize and ensure orientation preservation for relatively complex tasks. Furthermore, after extracting continuous features in the initial layers of CpAE (layers 1-3 in our experiments), the encoder processes the features through several subsequent CNN layers (layers 4-8 in our experiments) to derive the final latent states. These later CNN layers are Lipschitz continuous functions, which impact the continuity of the final latent variables, as reflected by the constant $C$ in Theorem 3.1. Numerically, this regularizer also serves as a latent similarity loss, helping to penalize the large constant $C$ in Theorem 3.1.
>
>   To address your concern, we have provided an explanation of this point in Section 4, page 8, of the revised manuscript.
>
>
> - **Effect of large filters**.
>
>   Larger filters are essential to ensure continuity. For a filter in the $l$-th layer of size $J_l \times J_l$, let $m_l = \max_{j_1, j_2} |W_l(j_1\delta, j_2\delta)|$ represent the largest weight value. The distance from the location of this maximum value to the filter boundary must be less than $\lceil J_l/2 \rceil \delta$. This constraint leads to $\max_l m_l / (\lceil J_l/2 \rceil \delta) \leq c_{\mathcal{W}}$. Therefore, using filters of smaller size can result in the failure of regularization ( $c_{\mathcal{W}}$ is very large) or cause all weights to converge to values close to zero ( $\max_l m_l$ is very small). To address your concern, we have provided an explanation of this point in Section 3.5, page 7, of the revised manuscript.
>
> Once again, we are grateful for the careful reading and valuable comments. We hope we have responded to all in an acceptable way and believe that the paper is quite improved because of this revision.

---

> > ### Comment · Reviewer_UE7U · 2024-11-29
> >
> > With consideration of the authors' responses and the comments of other reviewers I have increased my score.

---

### Official Review · Reviewer_5dSC · 2024-11-08

**Soundness:** 3
**Presentation:** 3
**Contribution:** 3
**Rating:** 6
**Confidence:** 3

**Summary:**

This paper introduces Continuity-Preserving Autoencoders (CpAEs), aimed at learning continuous latent dynamical models from discrete image data. It addresses the challenge of modeling continuous dynamical systems from image trajectories by proposing a new autoencoder architecture that promotes continuity in the latent space. CpAEs are designed to maintain continuous evolution in latent states by enforcing Lipschitz continuity in convolutional filters, with experiments demonstrating improved performance over standard methods, particularly for complex dynamical systems.

**Strengths:**

* The paper identifies an important problem: standard autoencoders struggle to learn continuous latent states from discrete image observations of dynamical system.
* The paper provides rigorous mathematical formulation and theoretical analysis, and a key theorem (theorem 3.1) that establishes conditions for continuous latent state evolution.
* Empirical results demonstrate the effectiveness across multiple datasets and clear improvements over baseline methods.

**Weaknesses:**

* Theoretically, the analysis is limited to rigid body motion in 2D plane and does not fully address non-rigid motion cases. The analysis assumes simple transformations (translation and rotation) that may not capture real-world complexity.
* The Assumption 3.1 about vanishing feature maps near image boundaries may be too restrictive, which may requires significant zero padding.
* The requirement for large filter sizes (O($1/\delta$)) could limit practical applications.
* The significant drop in performance on the swing stick task (57.4% VPT vs 99.2% for damped pendulum) is not thoroughly analyzed.

**Questions:**

Would the method work on more complicated mechanical system beyond rigid bodies?

---

> ### Author Response · Authors · 2024-11-21
> **Response to Reviewer 5dSC (1/2)**
>
> Thank you for your insightful feedback on our manuscript. Please find below a detailed response.
>
> - **Analysis is limited to rigid body motion in the 2D plane.**
>
>   We acknowledge that our current analysis primarily focuses on rigid body motion in a 2D plane. However, the mathematical formulation presented in Section 3.1 and the corresponding experiments are not limited to rigid body motion. In addition, considering 2D motion is natural in our context, as the dynamics are captured through 2D images.
>
>   Rigid body motion serves as a foundational case for our analysis, enabling us to establish a clear framework before tackling more complex scenarios. Specifically, the position of a rigid body can be succinctly described by translation and rotation, which facilitates a straightforward representation of the mapping $\mathcal{S}$ introduced in Section 3.1 and elaborated upon in Appendix A.1. Leveraging this representation, we rigorously prove Theorem 3.1.
>
>   In contrast, general motion encompasses a much broader range of possibilities, which complicates the explicit representation of the mapping $\mathcal{S}$. We believe that in practical applications, it may be necessary to tailor the representation of $\mathcal{S}$ to the specific characteristics of the motion under consideration.
>
>   To address your concern, in our revised manuscript, Section 5, page 10, we have added a discussion to clearly state the limitations of our current approach and outline our future research directions to address non-rigid motion cases.
>
> - **Performance of our method on systems beyond rigid bodies.**
>
>   Although our analysis (Theorem 3.1) is focused on rigid body motion, our method demonstrates strong performance even on non-rigid motion.
>
>   In Section 4.2 of the revised manuscript, we numerically verify its effectiveness on non-rigid motion using the elastic double pendulum, where each pendulum arm can stretch and contract. The results consistently show improved performance compared to baseline methods.
>
>   To address your concern, we have mentioned this point in Section 4.2, page 9, of the revised manuscript.
>
> - **Assumption 3.1 may be too restrictive.**
>
>   Thanks for the comments. We acknowledge that Assumption 3.1 might appear restrictive for general image tasks. However, in the context of learning dynamical systems from images, Assumption 3.1 is both mild and easily satisfied for scenarios where the objects of interest are well captured and typically located within the central region of the image. This setup naturally results in many zeros near the boundaries (i.e., the background of the images). Many practical applications, including all our numerical experiments, conform to this scenario.
>
>   To address this concern, we have included a discussion in Section 3.4, page 6, of the revised manuscript.

---

> ### Author Response · Authors · 2024-11-21
> **Response to Reviewer 5dSC(2/2)**
>
> - **Requirement for large filter sizes.**
>
>   We acknowledge that large filter sizes are a necessary condition to achieve the desired continuity guarantees as discussed in Section 3.5. Specifically, this requirement stems from the inequality $\max_l \max_{j_1, j_2} | W_l (j_1 \delta, j_2 \delta) | / (\lceil J_l/2 \rceil \delta) \leq c_{\mathcal{W}}$. To address this, we have further refined our conclusions to mitigate the need for overly large filters. In Theorem 3.1, we demonstrate that the constant of the translation component is ${c_{\mathcal{W}}}/{2^{L^*-1}}$. Assuming $\frac{c_{\mathcal{W}}}{2^{L^*-1}}, \max_l \max_{j_1, j_2} |W_l(j_1\delta, j_2\delta)| = \mathcal{O}(1)$, it follows that we only need to ensure $2^{L^*} J_l = \mathcal{O}(1/\delta)$. Consequently, in our experiments, we set $L^* = 4$ and applied a filter size of $12$ in the first three layers.
>
>   We also recognize that using larger filters increases computational costs in terms of memory and processing time. However, this increase is marginal and does not pose any fundamental challenges. All our experiments, including both the baseline and our method, were conducted on a single NVIDIA 3090 GPU. We believe the additional computational resources required are entirely justified given the significant performance improvements achieved.
>
> - **Analysing the drop in performance on the swing stick task**.
>
>   In our revised manuscript, Section 5, page 10, we have included a discussion of the factors contributing to this performance discrepancy. Key aspects we have considered include:
>   - Task Complexity: For the damped pendulum task, the underlying system is well understood, allowing us to ensure that the data spans the entire phase space. In contrast, the swing stick task involves a more complex dynamical system with higher degrees of freedom and real-world dynamics that are not fully characterized. This makes it challenging to collect data that adequately covers all necessary conditions.
>   - Data Quality: The damped pendulum data is generated through simulations, resulting in a clean, noise-free dataset with a simple background. On the other hand, the swing stick task data is obtained from recordings of real-world dynamics, which inherently include a complex background and significant noise.
>
> Once again, we are grateful for the careful reading and valuable comments. We hope we have responded to all in an acceptable way and believe that the paper is quite improved because of this revision.

---

> > ### Comment · Reviewer_5dSC · 2024-11-26
> >
> > Thanks for the authors' thorough reply. My most concerns have been addressed. I'll keep my score as positive.

---

### Author Response · Authors · 2024-11-21
**General Response**

We would like to express our sincere gratitude to the reviewers for their constructive comments and valuable suggestions. We have carefully considered each point raised and made corresponding revisions to the manuscript. In particular, we added additional numerical results to illustrate the following aspects:

- Neither the standard autoencoder using large filters alone nor the addition of a conventional $L_2$ regularizer can extract continuously evolving latent states. In contrast, the proposed continuity regularizer ensures continuous latent state evolution (see Section 4.1, page 8).

- Conventional CNN-based autoencoders fail to achieve $\delta$-continuity, which leads to the failure of subsequent Neural ODE training. In contrast, the continuous latent states learned by CpAEs enable the Neural ODE to effectively capture their dynamics (see Section 4.1, page 8).

- The regularization weight hyperparameters control the trade-off between model complexity and the continuity of the filter (see Appendix A.4.2, pages 23–24).

We hope that our responses have addressed all the concerns from the reviewers, and the revised manuscript is clearer as a result. The new and/or revised text in the revised manuscript is colored in blue. Please see the revised manuscript alongside the responses below.

If there are any concerns, uncertainties, or questions, we are happy to address them and provide further clarifications during the discussion phase. We look forward to further communication.

---

### Meta-Review · Area_Chair_JGwL · 2024-12-18

**Metareview:**

The paper presents a method for learning underlying continuous dynamics which generate complex high dimensional data such as images. This is done though an autoencoder method with restrictions on its architecture which encourage the learning of desirable representations.  The authors provide empirical results which demonstrate the utility of the method in a variety of relevant settings. Weaknesses of the work include the possibility that the architectural restrictions may not mesh well with more favored architectures in today's machine learning research. Overall reviewers had favorable opinions of the work and I will recommend acceptance.

**Additional Comments On Reviewer Discussion:**

Initial reviews were mixed/leaning-positive and reviewers bought up similar issues. They discussed theoretical details, metrics used (such as using reconstruction as an indicator of latent continuity), requests for additional experiments, architectural constraints, and some aberrant results.

The authors responded in detail. They added new experiments, ablations, explanation, and acknowledged some additional limitations in the method. As well, they expanded upon the analysis initially presented in the work.

The authors were very responsive to reviewer feedback and made a number of improvements to the paper, leading to increased scores from multiple reviewers.

The final version appears stronger, though some fundamental limitations (like CNN-only focus) remain as areas for future work.

---

### Decision · Program_Chairs · 2025-01-22

Accept (Poster)